# Deployment-Efficient Reinforcement Learning via Model-Based Offline Optimization

**Tatsuya Matsushima**[*]  **Hiroki Furuta**[*]  **Yutaka Matsuo**
The University of Tokyo
{matsushima, furuta, matsuo}@weblab.t.u-tokyo.ac.jp

**Ofir Nachum**  **Shixiang Shane Gu**
Google Research
{ofirnachum, shanegu}@google.com

## Abstract

Most reinforcement learning (RL) algorithms assume online access to the environment, in which one may readily interleave updates to the policy with experience collection using that policy. However, in many real-world applications such as health, education, dialogue agents, and robotics, the cost or potential risk of deploying a new data-collection policy is high, to the point that it can become prohibitive to update the data-collection policy more than a few times during learning. With this view, we propose a novel concept of *deployment efficiency*, measuring the number of distinct data-collection policies that are used during policy learning. We observe that naïvely applying existing model-free offline RL algorithms recursively does not lead to a practical deployment-efficient *and* sample-efficient algorithm. We propose a novel model-based algorithm, Behavior-Regularized Model-ENsemble (BREMEN), that not only performs better than or comparably as the state-of-the-art dynamic-programming-based and concurrently-proposed model-based offline approaches on existing benchmarks, but can also effectively optimize a policy offline using 10-20 times fewer data than prior works. Furthermore, the recursive application of BREMEN achieves impressive deployment efficiency while maintaining the same or better sample efficiency, learning successful policies from scratch on simulated robotic environments with only 5-10 deployments, compared to typical values of hundreds to millions in standard RL baselines. [1]

## 1 Introduction

Reinforcement learning (RL) algorithms have recently demonstrated impressive success in learning behaviors for a variety of sequential decision-making tasks (Barth-Maron et al., 2018; Hessel et al., 2018; Nachum et al., 2019). Virtually all of these demonstrations have relied on highly-frequent online access to the environment, with the RL algorithms often interleaving each update to the policy with additional experience collection of that policy acting in the environment. However, in many real-world applications of RL, such as health (Murphy et al., 2001), education (Mandel et al., 2014), dialog agents (Jaques et al., 2019), and robotics (Gu et al., 2017a; Kalashnikov et al., 2018), the deployment of a new data-collection policy may be associated with a number of costs and risks. If we can learn tasks with a small number of data collection policies, we can substantially reduce them.

Based on this idea, we propose a novel measure of RL algorithm performance, namely *deployment efficiency*, which counts the number of changes in the data-collection policy during learning, as illustrated in Figure 1. This concept may be seen in contrast to *sample efficiency* or *data efficiency* (Precup et al., 2001; Degris et al., 2012; Gu et al., 2017b; Haarnoja et al., 2018; Lillicrap et al., 2016; Nachum et al., 2018), which measures the amount of environment interactions incurred during training, without regard to how many distinct policies were deployed to perform those interactions. Even when the

---

[*]Equal contribution.
[1]Codes and pre-trained models are available at https://github.com/matsuolab/BREMEN.

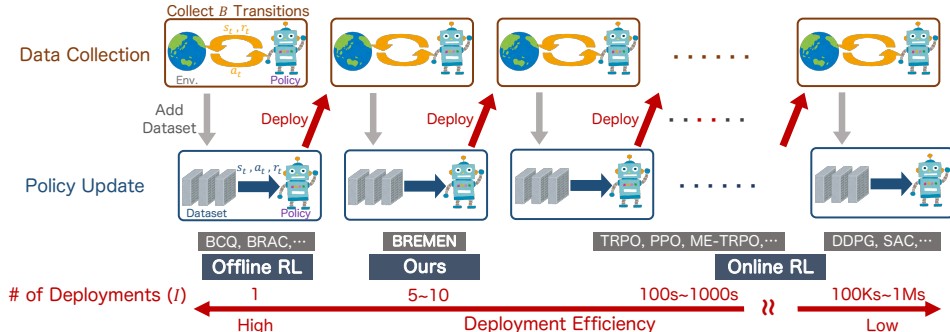

Figure 1: *Deployment efficiency* is defined as the number of changes in the data-collection policy ($I$), which is vital for managing costs and risks of new policy deployment. Online RL algorithms typically require many iterations of policy deployment and data collection, which leads to extremely low deployment efficiency. In contrast, most pure offline algorithms consider updating a policy from a fixed dataset without additional deployment and often fail to learn from a randomly initialized data-collection policy. Interestingly, most state-of-the-art off-policy algorithms are still evaluated in heavily online settings. For example, SAC (Haarnoja et al., 2018) collects one sample per policy update, amounting to 100,000 to 1 million deployments for learning standard benchmark domains.

data efficiency is high, the deployment efficiency could be low, since many on-policy and off-policy algorithms alternate data collection with each policy update (Schulman et al., 2015; Lillicrap et al., 2016; Gu et al., 2016; Haarnoja et al., 2018). Such dependence on high-frequency policy deployments is best illustrated in the recent works in offline RL (Fujimoto et al., 2019; Jaques et al., 2019; Kumar et al., 2019; Levine et al., 2020; Wu et al., 2019), where baseline off-policy algorithms exhibited poor performance when trained on a static dataset. These offline RL works, however, limit their study to a single deployment, which is enough for achieving high performance with data collected from a sub-optimal behavior policy, but often not from a random policy. In contrast to those prior works, we aim to learn successful policies from scratch in a manner that is both sample and deployment-efficient.

Many existing model-free offline RL algorithms (Levine et al., 2020) are tuned and evaluated on massive datasets (e.g., one million transitions). In order to develop an algorithm that is both sample and deployment-efficient, each iteration of the algorithm between successive deployments has to work effectively on much smaller dataset sizes. We believe model-based RL is better suited to this setting due to its higher demonstrated sample efficiency than model-free RL (Kurutach et al., 2018; Nagabandi et al., 2018). Although the combination of model-based RL and offline or limited-deployment settings seems straight-forward, we find this naïve approach leads to poor performance. This problem can be attributed to *extrapolation errors* (Fujimoto et al., 2019) similar to those observed in model-free methods. Specifically, the learned policy may choose sequences of actions which lead it to regions of the state space where the dynamics model cannot predict properly, due to poor coverage of the dataset. This can lead the policy to exploit approximation errors of the dynamics model and be disastrous for learning. In model-free settings, similar data distribution shift problems are typically remedied by regularizing policy updates explicitly with a divergence from the observed data distribution (Jaques et al., 2019; Kumar et al., 2019; Wu et al., 2019), which, however, can overly limit policies' expressivity (Sohn et al., 2020).

In order to better approach these problems arising in limited deployment settings, we propose Behavior-Regularized Model-ENsemble (BREMEN), which learns an ensemble of dynamics models in conjunction with a policy using imaginary rollouts while *implicitly* regularizing the learned policy via appropriate parameter initialization and conservative trust-region learning updates. We evaluate BREMEN on standard offline RL benchmarks of high-dimensional continuous control tasks, where only a single static dataset is used. In this fixed-batch setting, our experiments show that BREMEN can not only achieve performance competitive with state-of-the-art when using standard dataset sizes but also learn with 10-20 times smaller datasets, which previous methods are unable to attain. Enabled by such stable and sample-efficient offline learning, we show that BREMEN can learn successful policies with only 5-10 deployments in the online setting, significantly outperforming existing off-policy and offline RL algorithms in deployment efficiency while keeping sample efficiency.

## 2  PRELIMINARIES

We consider a Markov Decision Process (MDP) setting, characterized by the tuple $\mathcal{M} = (\mathcal{S}, \mathcal{A}, p, r, \gamma)$, where $\mathcal{S}$ is the state space, $\mathcal{A}$ is the action space, $p(s'|s, a)$ is the transition probability distribution or dynamics, $r(s)$ is the reward function and $\gamma \in (0, 1)$ is the discount factor. A policy $\pi$ is a function that determines the agent behavior, mapping from states to probability distributions over actions. The goal is to obtain the optimal policy $\pi^*$, which maximizes the expectation of discounted sum of rewards. The transition probability $p(s'|s, a)$ is usually unknown, and estimated with a parameterized dynamics model $f_\phi$ (e.g. a neural network) in model-based RL. For simplicity, we assume that the reward function $r(s)$ is known, and the reward can be computed for any arbitrary state, but we may extend to the unknown setting and predict it using a parameterized function.

**On-policy vs Off-policy, Online vs Offline** At a high-level, most RL algorithms alternate many times between collecting a batch of transitions (deployments) and optimizing the policy (learning). If the algorithms discard data after each policy update, they are *on-policy* (Schulman et al., 2015; 2017), while if they accumulate data in a buffer $\mathcal{D}$, i.e. experience replay (Lin, 1992), they are *off-policy* (Mnih et al., 2015; Lillicrap et al., 2016; Gu et al., 2016; 2017b; Haarnoja et al., 2018; Fujimoto et al., 2019; Fakoor et al., 2019) because not all the data in buffer comes from the current policy. However, we consider all these algorithms to be *online* RL algorithms, since they involve many deployments during learning, ranging from hundreds to millions. On the other hand, in pure *offline* RL, one does not assume direct interaction and learns a policy from only a fixed dataset, which effectively corresponds to a single deployment allowed for learning. Classically, interpolating these two extremes were semi-batch RL algorithms (Lange et al., 2012; Singh et al., 1995), which improve the policy through repetitions of collecting a large batch of transitions $\mathcal{D} = \{(s, a, s', r)\}$ and performing many or full policy updates. While these semi-batch RL also realize good deployment efficiency, they have not been extensively studied with neural network function approximators or in off-policy settings with experience replay for scalable sample-efficient learning. In our work, we aim to have both high deployment efficiency and sample efficiency by developing an algorithm that can solve the tasks with minimal policy deployments as well as transition samples.

## 3  DEPLOYMENT EFFICIENCY

Deploying a new policy for data collection can be associated with a number of costs and risks for many real-world applications like medicine, dialogue systems, or robotic control (Murphy et al., 2001; Mandel et al., 2014; Gu et al., 2017a; Kalashnikov et al., 2018; Nachum et al., 2019; Jaques et al., 2019). While there are abundant works on safety for RL (Chow et al., 2015; Eysenbach et al., 2018; Chow et al., 2018; Ray et al., 2019; Chow et al., 2019), they often do not provide guarantees in practice when combined with neural networks and stochastic optimization. It is therefore necessary to verify each policy before deployment (e.g. measuring the variance of rewards or checking out-of-bounds actions). Due to such costs associated with each deployment, it is desirable to minimize the number of distinct deployments needed during the learning process. Even ignoring safety considerations, frequent updates to a deployed policy can exacerbate communication bottlenecks in large-scale distributed RL systems, which are becoming more prevalent (Nair et al., 2015; Espeholt et al., 2018; 2019). We additionally discuss on the importance of the deployment efficiency in real-world applications. See Appendix C.

In order to focus research on these practical bottlenecks, we propose a novel measure of RL algorithms, namely, *deployment efficiency*, which counts how many times the data-collection policy has been changed during improvement from random policy to solve the task. For example, if an RL algorithm operates by using its learned policy to collect transitions from the environment $I$ times, each time collecting a batch of $B$ new transitions, then the number of deployments is $I$, while the total number of samples collected is $I \times B$. The lower $I$ is, the more deployment-efficient the algorithm is; in contrast, sample efficiency looks at $I \times B$. Online RL algorithms, whether they are on-policy or off-policy, typically update the policy and acquire new transitions by deploying the newly updated policy at every iteration. This corresponds to performing hundreds to millions of deployments during learning on standard benchmarks (Haarnoja et al., 2018), which is severely deployment inefficient. On the other hand, offline RL literature only studies the case of 1 deployment. A deployment-efficient algorithm would stand in the middle of these two extremes and ideally learn a successful policy from scratch while deploying only a few distinct policies, as illustrated in Figure 1.

Recent deep RL literature seldom emphasizes deployment efficiency, with few exceptions in specific applications (Kalashnikov et al., 2018) where such a learning procedure is necessary. Deployment-inefficient algorithms will fail in scenarios where the deployment of each new policy is exorbitantly expensive, such as safety-critical robotics or user-facing products. Although current state-of-the-art algorithms on continuous control have substantially improved sample or data efficiency, they have not optimized for deployment efficiency. For example, SAC (Haarnoja et al., 2018), an efficient model-free off-policy algorithm, performs half a million to one million policy deployments during learning on MuJoCo (Todorov et al., 2012) benchmarks. ME-TRPO (Kurutach et al., 2018), a model-based algorithm, performs a much lower 100-300 policy deployments, although this is still relatively high for practical settings.[2] In our work, we demonstrate successful learning on standard benchmark environments with only 5-10 deployments.

## 4 BEHAVIOR-REGULARIZED MODEL-ENSEMBLE

To achieve a favorable combination of both high deployment and sample efficiency, we propose Behavior-Regularized Model-ENsemble (BREMEN). BREMEN incorporates Dyna-style (Sutton, 1991; Kurutach et al., 2018) model-based RL, learning an ensemble of dynamics models in conjunction with a policy using imaginary rollouts and behavior regularization via conservative trust-region updates.

### 4.1 IMAGINARY ROLLOUT FROM MODEL ENSEMBLE

As in recent Dyna-style model-based RL methods (Kurutach et al., 2018; Wang et al., 2019), BREMEN uses an ensemble of $K$ deterministic dynamics models $\hat{f}_\phi = \left\{ \hat{f}_{\phi_1}, \ldots, \hat{f}_{\phi_K} \right\}$ to alleviate the problem of model bias. Each model $\hat{f}_{\phi_i}$ is parameterized by $\phi_i$ and trained by the following objective, which minimizes mean squared error between the prediction of next state $\hat{f}_{\phi_i}(s_t, a_t)$ and true next state $s_{t+1}$ over a dataset $\mathcal{D}$:

$$\min_{\phi_i} \frac{1}{|\mathcal{D}|} \sum_{(s_t, a_t, s_{t+1}) \in \mathcal{D}} \frac{1}{2} \left\| s_{t+1} - \hat{f}_{\phi_i}(s_t, a_t) \right\|_2^2. \tag{1}$$

During training of a policy $\pi_\theta$, imagined trajectories of states and actions are generated sequentially, using a dynamics model $\hat{f}_{\phi_i}$ that is randomly selected at each time step:

$$a_t \sim \pi_\theta(\cdot|\hat{s}_t), \quad \hat{s}_{t+1} = \hat{f}_{\phi_i}(\hat{s}_t, a_t) \quad \text{where} \quad i \sim \{1 \cdots K\}. \tag{2}$$

### 4.2 POLICY UPDATE WITH BEHAVIOR REGULARIZATION

In order to manage the discrepancy between the true dynamics and the learned model caused by the distribution shift in batch settings, we propose to use iterative policy updates via a trust-region constraint, re-initialized with a behavior-cloned policy after every deployment. Specifically, after each deployment, we are given an updated dataset of experience transitions $\mathcal{D}$. With this dataset, we approximate the true behavior policy $\pi_b$ through behavior cloning (BC), utilizing a neural network $\hat{\pi}_\beta$ parameterized by $\beta$, where we implicitly assume a fixed variance, a common practice in BC (Rajeswaran et al., 2017):

$$\min_{\beta} \frac{1}{|\mathcal{D}|} \sum_{(s_t, a_t) \in \mathcal{D}} \frac{1}{2} \|a_t - \hat{\pi}_\beta(s_t)\|_2^2. \tag{3}$$

After obtaining the estimated behavior policy, we initialize the target policy $\pi_\theta$ as a Gaussian policy with mean from $\hat{\pi}_\beta$ and standard deviation of $1$. This BC initialization in conjunction with gradient descent based optimization may be seen as implicitly biasing the optimized $\pi_\theta$ to be close to the data-collection policy (Nagarajan & Kolter, 2019), and thus works as a remedy for the distribution shift problem (Ross et al., 2011). To further bias the learned policy to be close to the data-collection

---

[2]We examined the number of deployments by checking their original implementations, while the frequency of data collection is a tunable hyper-parameter.

---

**Algorithm 1** BREMEN for Deployment-Efficient RL

---

**Input:** Empty dataset $\mathcal{D}_{all}$, $\mathcal{D}$, Initial parameters $\phi = \{\phi_1, \cdots, \phi_K\}$, $\beta$, Number of policy optimization $T$, Number of deployments $I$.
1: Randomly initialize the target policy $\pi_\theta$.
2: **for** deployment $i = 1, \cdots, I$ **do**
3:     Collect $B$ transitions in the true environment using $\pi_\theta$ and add them to dataset
        $\mathcal{D}_{all} \leftarrow \mathcal{D}_{all} \cup \{s_t, a_t, r_t, s_{t+1}\}, \mathcal{D} \leftarrow \{s_t, a_t, r_t, s_{t+1}\}$.
4:     Train $K$ dynamics models $\hat{f}_\phi$ using $\mathcal{D}_{all}$ via Equation 1.
5:     Train estimated behavior policy $\hat{\pi}_\beta$ using $\mathcal{D}$ by behavior cloning via Equation 3.
6:     Re-initialize target policy $\pi_{\theta_0} = \text{Normal}(\hat{\pi}_\beta, 1)$.
7:     **for** policy optimization $k = 1, \cdots, T$ **do**
8:         Generate imaginary rollout via Equation 2.
9:         Optimize target policy $\pi_\theta$ satisfying Equation 4 with the rollout.

---

policy, we opt to use a KL-based trust-region optimization (Schulman et al., 2015). Therefore, the optimization of BREMEN becomes

$$\theta_{k+1} = \arg\max_\theta \mathbb{E}_{s,a \sim \pi_{\theta_k}, \hat{f}_{\phi_i}} \left[ \frac{\pi_\theta(a|s)}{\pi_{\theta_k}(a|s)} A^{\pi_{\theta_k}}(s, a) \right] \tag{4}$$
$$\text{s.t.} \quad \mathbb{E}_{s \sim \pi_{\theta_k}, \hat{f}_{\phi_i}} \left[ D_{\text{KL}} \left( \pi_\theta(\cdot|s) \| \pi_{\theta_k}(\cdot|s) \right) \right] \leq \delta, \quad \pi_{\theta_0} = \text{Normal}(\hat{\pi}_\beta, 1),$$

where $A^{\pi_{\theta_k}}(s, a)$ is the advantage of $\pi_{\theta_k}$ computed using model-based rollouts in the learned dynamics model and $\delta$ is the maximum step size.

The combination of BC for initialization and finite iterative trust-region updates serves as an implicit KL regularization. This is in contrast to many previous offline RL algorithms that augment the value function with a penalty of explicit KL divergence (Siegel et al., 2020; Wu et al., 2019) or maximum mean discrepancy (Kumar et al., 2019). Empirically, we found that our regularization technique outperforms the explicit KL penalty (Section 5.3). Furthermore, we provide a mathematical intuition explaining how our methods works as an implicit regularization of distributional shift in Appendix A.

By recursively performing offline procedure, BREMEN can be used for deployment-efficient learning as shown in Algorithm 1, starting from a randomly initialized policy, collecting experience data, and performing offline policy updates.

## 5 EXPERIMENTS

In order to realize a deployment-efficient RL algorithm, the batch policy optimizer has to be stable and sample-efficient. We first evaluate BREMEN in the offline setting, where the algorithm learns the policy from a static dataset. Standard benchmarks of MuJoCo physics simulator shown in (Wu et al., 2019) and more recent datasets (Fu et al., 2020) are used in the evaluation, and we compared the asymptotic performance of BREMEN with other offline RL methods including the concurrent model-based approaches. We then tested the sample-efficiency of offline algorithms using smaller datasets. We lastly extend the experiment to deployment-efficient settings, where the algorithms learn their policies from scratch via a limited number of deployments and perform some ablations to see how components in BREMEN affect performance. See Appendix F for further details.

### 5.1 EVALUATING OFFLINE RL PERFORMANCES

**Standard Benchmarks** We evaluate BREMEN on standard offline RL benchmarks following and identical protocol as in Wu et al. (2019): We first train online SAC to a certain cumulative reward threshold, 4,000 in HalfCheetah, 1,000 in Ant, Hopper, and Walker2d, and collect offline datasets. We evaluate agents with the offline dataset of one million (1M) transitions, which is standard for BCQ and BRAC. Table 1 (top) shows that BREMEN can achieve performance competitive with state-of-the-art model-free offline RL algorithms when using the standard dataset size of 1M. We also test BREMEN with more recent benchmarks of D4RL (Fu et al., 2020) and compared the performance with the existing model-free and model-based methods. See Appendix D for the results.

| 1,000,000 (1M) transitions | | | | |
|---|---|---|---|---|
| **Method** | **Ant** | **HalfCheetah** | **Hopper** | **Walker2d** |
| Dataset | 1191 | 4126 | 1128 | 1376 |
| BC | 1321±141 | 4281±12 | 1341±161 | 1421±147 |
| BCQ | 2021±31 | 5783±272 | 1130±127 | 2153±753 |
| BRAC | 2072±285 | 7192±115 | 1422±90 | 2239±1124 |
| BRAC (max Q) | 2369±234 | 7320±91 | 1916±343 | **2409±1210** |
| BREMEN (Ours) | **3328±275** | **8055±103** | **2058±852** | 2346±230 |
| ME-TRPO (offline) | 1258±550 | 1804±924 | 518±91 | 211±154 |
| 100,000 (100K) transitions | | | | |
| **Method** | **Ant** | **HalfCheetah** | **Hopper** | **Walker2d** |
| Dataset | 1191 | 4066 | 1128 | 1376 |
| BC | 1330±81 | 4266±21 | 1322±109 | 1426±47 |
| BCQ | 1363±199 | 3915±411 | 1129±238 | **2187±196** |
| BRAC | -157±383 | 2505±2501 | 1310±70 | 2162±1109 |
| BRAC (max Q) | -226±387 | 2332±2422 | 1422±101 | 2164±1114 |
| BREMEN (Ours) | **1633±127** | **6095±370** | **2191±455** | 2132±301 |
| ME-TRPO (offline) | 974±4 | 2±434 | 307±170 | 10±61 |
| 50,000 (50K) transitions | | | | |
| **Method** | **Ant** | **HalfCheetah** | **Hopper** | **Walker2d** |
| Dataset | 1191 | 4138 | 1128 | 1376 |
| BC | 1270±65 | 4230±49 | 1249±61 | 1420±194 |
| BCQ | 1329±95 | 1319±626 | 1178±235 | 1841±439 |
| BRAC | -878±244 | -597±73 | 1277±102 | 976±1207 |
| BRAC (max Q) | -843±279 | -590±56 | 1276±225 | 903±1137 |
| BREMEN (Ours) | **1347±283** | **5823±146** | **1632±796** | **2280±647** |
| ME-TRPO (offline) | 938±32 | -73±95 | 152±13 | 176±343 |

Table 1: Comparison of BREMEN to the existing offline methods on static datasets. Each cell shows the average cumulative reward and their standard deviation, where the number of samples is 1M, 100K, and 50K, respectively. The maximum steps per episode is 1,000. BRAC applies a primal form of KL value penalty, and BRAC (max Q) means its variant of sampling multiple actions and taking the maximum according to the learned Q function.

**Evaluating Sample-Efficiency** We then evaluate the sample-efficiency by making much smaller datasets of 50k and 100k transitions (5∼10 % of Wu et al. (2019)). Surprisingly, Table 1 (middle and bottom) shows that BREMEN can also learn with smaller datasets, where BCQ and BRAC are unable to exceed even BC baseline. This is a novel evaluation protocol we proposed, and our BREMEN's superior performance here is exactly what enables recursive BREMEN in the next section to be an effective algorithm in deployment-constrained settings.

## 5.2 EVALUATING DEPLOYMENT EFFICIENCY IN ONLINE RL BENCHMARKS

We compare BREMEN to ME-TRPO, SAC, BCQ, and BRAC applied to limited deployment settings. To adapt offline methods (BCQ, BRAC) to this setting, we simply apply them in a recursive fashion;[3] at each deployment iteration, we collect a batch of data with the most recent policy and then run the offline update with this dataset. As for SAC, we simply change the replay buffer to update only at specific deployment intervals. For the sake of comparison, we align the number of deployments and the amount of data collection at each deployment (either 100k or 200k) for all methods.[4]

Figure 2 shows the results with 200k (top) and 100k (bottom) batched transitions per deployment. Regardless of the environments and the batch size per update, BREMEN achieves remarkable performance while existing online and offline RL methods struggle to make any progress. As a point of comparison, we also include results for online SAC and ME-TRPO without deployment-limits but using the same number of transitions. We additionally compare BREMEN to the model-based offline RL methods with uncertainty-based penalties. See Appendix E for further details.

Following the motivation of deployment efficiency, obtaining a successful policy under data-collection constraint conditions in the real application, we extensively evaluate our algorithm on more realistic robotics environments in OpenAI Gym. The experimental procedure is the same as above, while

---

[3]Recursive BCQ and BRAC also do behavioral cloning-based policy initialization after each deployment.
[4]We evaluate the trade-off between sample and deployment efficiency in Appendix B.

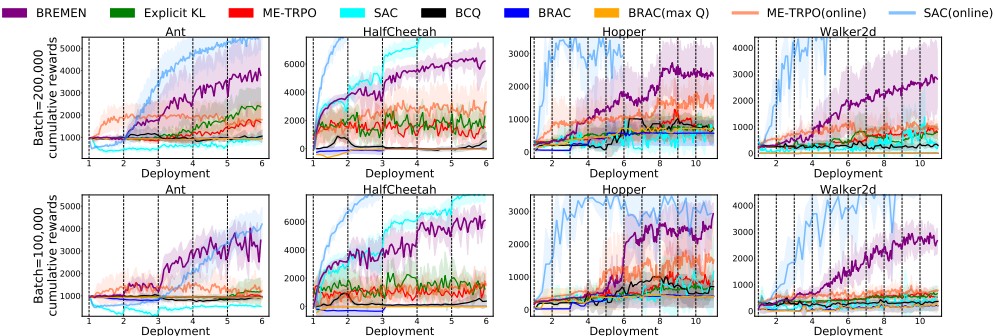

Figure 2: Evaluation of BREMEN with the existing methods (ME-TRPO, SAC, BCQ, BRAC) under deployment constraints (to 5-10 deployments with batch sizes of 200k and 100k). The average cumulative rewards and their standard deviations with 5 random seeds are shown. Vertical dotted lines represent where each policy deployment and data collection happen. BREMEN is able to learn successful policies with only 5-10 deployments, while the state-of-the-art off-policy (SAC), model-based (ME-TRPO), and recursively-applied offline RL algorithms (BCQ, BRAC) often struggle to make any progress. For completeness, we show ME-TRPO(online) and SAC(online) which are their original optimal learning curves without deployment constraints, plotted with respect to samples normalized by the batch size. While SAC(online) substantially outperforms BREMEN in sample efficiency, it uses 1 deployment per sample, leading to 100k-500k deployments required for learning. Interestingly, BREMEN achieves even better performance than the original ME-TRPO(online), suggesting the effectiveness of implicit behavior regularization. For SAC and ME-TRPO under deployment-constrained evaluation, their batch size between policy deployments differs substantially from their standard settings, and therefore we performed extensive hyper-parameter search on the relevant parameters such as the number of policy updates between deployments, as discussed in Appendix F.2.1.

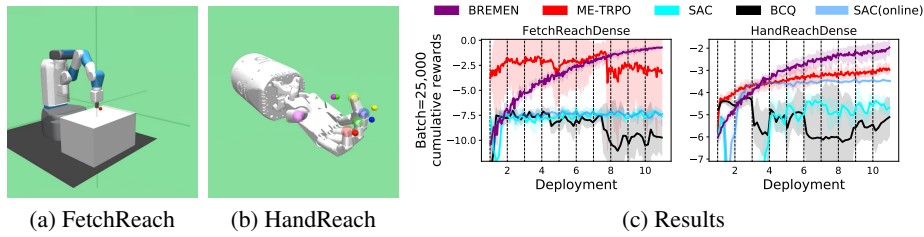

(a) FetchReach     (b) HandReach            (c) Results

Figure 3: Robotics environments and the results under deployment constraints (10 deployments with batch sizes of 25k). The performances are averaged over 5 seeds. BREMEN seems the only method that shows both stable improvement and solving tasks without large degradation or sub-optimal convergence.

we limit the batch size at each deployment as only 25k. Figure 3 presents the reaching tasks with Fetch robot and 20-DoF shadow hand (Plappert et al., 2018), and the experimental results in both environments. Only BREMEN shows stable improvement and high performance, while other offline and online algorithms fail to learn. These results suggest that a model-based method is a desirable approach for satisfying practical requirements in robotics, i.e. sample and deployment efficiency.

## 5.3 ABLATION: EVALUATING EFFECTIVENESS OF IMPLICIT KL CONTROL

In this section, we present an experiment to better understand the effect of BREMEN's implicit regularization. Figure 4 shows the KL divergence of learned policies from the last deployed policy. We compare BREMEN to variants of BREMEN that use an explicit KL penalty on value instead of BC initialization (conservative KL trust-region updates are still used). We find that the explicit KL without behavior initialization variants learn policies that move farther away from the last deployed policy than behavior initialized policies. This suggests that the implicit behavior regularization employed by BREMEN is more effective as a conservative policy learning protocol. In addition, to assess the effect of repeated behavior cloning initialization, we also evaluate a variant of BREMEN without behavior cloning re-initialization (grey). This variant works in easier environments (Ant, Halfcheetah), but does not show remarkable progress in more challenging ones with termination (Hopper, Walker2d).

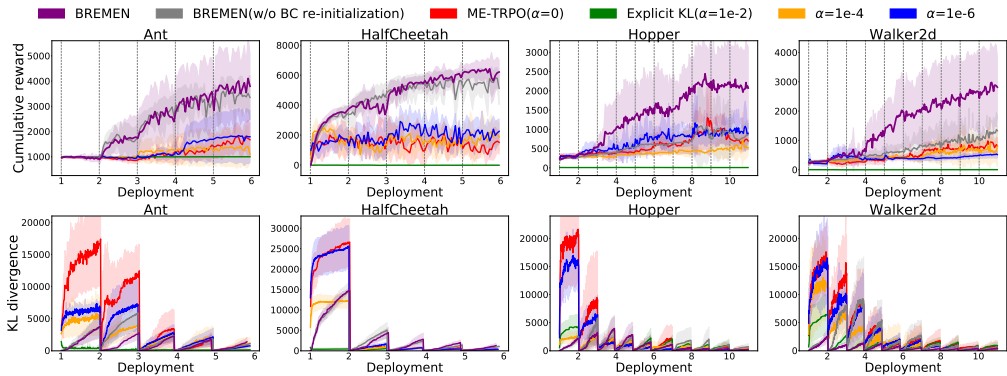

Figure 4: We examine average cumulative rewards (top) and corresponding KL divergence between the last deployed policy and the target policy (bottom) with batch size 200K in limited deployment settings. The behavior initialized policy remains close to the last deployed policy during improvement without explicit value penalty $-\alpha D_{\mathrm{KL}}(\pi_\theta \| \hat{\pi}_\beta)$. The explicit penalty is controlled by a coefficient $\alpha$.

This result empirically supports the need for repeated behavior initialization after each deployment. The results of further experiments are shown in Appendix G.

## 6 RELATED WORK

**Deployment Efficiency and Offline RL** Although we are not aware of any previous works which explicitly proposed the concept of deployment efficiency, its necessity in many real-world applications has been generally known. One may consider previously proposed semi-batch RL algorithms (Ernst et al., 2005; Lange et al., 2012; Singh et al., 1994; Roux, 2016) or theoretical analysis of switching cost under the tabular PAC-MDP settings (Bai et al., 2019; Guo & Brunskill, 2015) as approaching this issue. More recently, a related but distinct problem known as offline RL has gained popularity (Levine et al., 2020; Wu et al., 2019; Agarwal et al., 2019; Kumar et al., 2020). These works consider an extreme version of 1 deployment, and typically collect the static batch with a partially trained policy rather than a random policy. While offline RL has shown promising results for a variety of real-world applications, such as robotics (Mandlekar et al., 2019), dialogue systems (Jaques et al., 2019), or medical treatments (Gottesman et al., 2018), these algorithms struggle when learning a policy from scratch or when the dataset is small. Nevertheless, common themes of many offline RL algorithms – regularizing the learned policy to the behavior policy (Fujimoto et al., 2019; Kumar et al., 2019; Siegel et al., 2020; Wu et al., 2019) and utilizing ensembles to handle uncertainty (Kumar et al., 2019; Wu et al., 2019) – served as inspirations for the proposed our algorithm. A major difference of BREMEN from prior works is that the target policy is not explicitly forced to stick close to the estimated behavior policy through the policy update except for the initial iteration. Rather, BREMEN employs a more implicit regularization by initializing the learned policy with a behavior cloned policy and then applying conservative trust-region updates. Another major difference is the application of model-based approaches to fully offline settings, which has not been extensively studied in prior works (Levine et al., 2020), except the two concurrent works (Kidambi et al., 2020; Yu et al., 2020) that study pessimistic or uncertainty penalized MDPs with guarantees – closely related to Liu et al. (2019). By contrast, our work shows that a simple technique can already enable model-based offline algorithms to significantly outperform the prior model-free methods, and is, to the best of our knowledge, the first to define and extensively evaluate deployment efficiency with recursive experiments.

**Model-Based RL** There are many types of model-based RL algorithms (Sutton, 1991; Deisenroth & Rasmussen, 2011; Heess et al., 2015). A simple algorithmic choice is Dyna-style (Sutton, 1991), which uses a parameterized model to estimate the true MDP transition function, stochastically mapping states and actions to next states. The dynamics model can then serve as a simulator of the environment during policy updates. Dyna-style algorithms often suffer from the distributional shift, also known as model bias, which leads RL agents to exploit regions where the data is insufficient, and significant performance degradation. A variety of remedies been proposed to relieve the issue of model bias, such as the use of multiple dynamics models as an ensemble (Chua et al., 2018; Kurutach

et al., 2018; Janner et al., 2019), meta-learning (Clavera et al., 2018), energy-based regularizer (Boney et al., 2019), game-theoretic framework (Rajeswaran et al., 2020), and explicit penalty for unknown states (Kidambi et al., 2020; Yu et al., 2020). Notably, we have employed a subset of these remedies – model ensembles and trust-region updates (Kurutach et al., 2018) – for BREMEN. Compared to prior works, our work is notable for using BC initialization in conjunction with trust-region updates to alleviate the distribution shift of the learned policy from the dataset used to train the dynamics model.

## 7 CONCLUSION

In this work, we introduced *deployment efficiency*, a novel measure for RL performance that counts the number of changes in the data-collection policy during learning. To enhance deployment efficiency, we proposed a novel model-based offline algorithm, Behavior-Regularized Model-ENsemble (BRE-MEN), combining model-ensembles with trust region updates from model-based RL literature (Kuru-tach et al., 2018), and policy initialization with behavior cloning from offline RL literature (Fujimoto et al., 2019; Wu et al., 2019). Crucially, BREMEN can improve policies offline *sample-efficiently* even when the batch size is 10-20 times smaller than prior works, allowing BREMEN to achieve impressive results in limited deployment settings, obtaining successful policies from scratch in only 5-10 deployments. Not only can this help alleviate costs and risks in real-world applications, but it can also reduce the amount of communication required during distributed learning and could form the basis for communication-efficient large-scale RL in contrast to prior works (Nair et al., 2015; Espeholt et al., 2018; 2019). Most critically, we show that under deployment efficiency constraints, most prior algorithms – model-free or model-based, online or offline – *fail* to achieve successful learning. One possible direction for future work is to incorporate *verification efficiency* into consideration, since a stochastic multi-modal policy could collect more diverse transitions while it takes more trajectories to be *verified* for safety than a uni-modal policy. While we presented promising results on some realistic simulated environments, validating BREMEN on real robots is another direction. We hope our work can gear the research community to value deployment efficiency as an important criterion for RL algorithms, and to eventually achieve similar sample efficiency and asymptotic performance as the state-of-the-art algorithms like SAC (Haarnoja et al., 2018) while having the deployment efficiency well-suited for safe and practical real-world reinforcement learning.

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

APPENDIX

## A  IMPLICIT KL CONTROL FROM A MATHEMATICAL PERSPECTIVE

We can intuitively understand that behavior cloning initialization with trust-region updates works as a regularization of distributional shift, and this can be supported by theory. Following the notation of Janner et al. (2019), we denote the generalization error of a dynamics model on the state distribution under the true behavior policy as $\epsilon_m = \max_t \mathbb{E}_{s \sim d_t^{\pi_b}} D_{TV}(p(s_{t+1}|s_t, a_t)||p_\phi(s_{t+1}|s_t, a_t))$, where $D_{TV}$ represents the total variation distance between true dynamics $p$ and learned model $p_\phi$. We also denote the distribution shift on the target policy as $\max_s D_{TV}(\pi_b||\pi) \leq \epsilon_\pi$. A bound relating the true returns $\eta[\pi]$ and the model returns $\hat{\eta}[\pi]$ on the target policy is given in Janner et al. (2019) as,

$$\eta[\pi] \geq \hat{\eta}[\pi] - \left[ \frac{2\gamma r_{max}(\epsilon_m + 2\epsilon_\pi)}{(1-\gamma)^2} + \frac{4r_{max}\epsilon_\pi}{(1-\gamma)} \right]. \tag{5}$$

This bound guarantees the improvement under the true returns as long as the improvement under the model returns increases by more than the slack in the bound due to $\epsilon_m, \epsilon_\pi$ (Janner et al., 2019; Levine et al., 2020).

We may relate this bound to the specific learning employed by BREMEN, which includes dynamics model learning, behavior cloning policy initialization, and conservative KL-based trust-region policy updates. To do so, we consider an *idealized* version of BREMEN, where the expectations over states in equations Equation 1, 3, 4 are replaced with supremums and the dynamics model is set to have unit variance.

**Proposition 1** (Policy and model error bound). *Suppose we apply the idealized BREMEN on a dataset $\mathcal{D}$, and define $\epsilon_\beta, \epsilon_\phi$ in terms of the behavior cloning and dynamics model losses as,*

$$\epsilon_\beta := \sup_s \mathbb{E}_{a \sim \mathcal{D}(-|s)}[\|a - \hat{\pi}_\beta(s)\|_2^2 / 2] - \mathcal{H}(\pi_b(-|s))$$

$$\epsilon_\phi := \sup_{s,a} \mathbb{E}_{s' \sim \mathcal{D}(-|s,a)} \left[ \|s' - \hat{f}_\phi(s,a)\|_2^2 / 2 \right] - \mathcal{H}(p(-|s,a)),$$

*where $\mathcal{H}$ denotes the Shannon entropy. If one then applies $T$ KL-based trust-region steps of step size $\delta$ (Equation 4) using stochastic dynamics models with mean $\hat{f}_\phi$ and standard deviation 1, then*

$$\epsilon_\pi = \sqrt{\frac{1}{2}\epsilon_\beta + \frac{d_a}{4}\log 2\pi} + T\sqrt{\frac{1}{2}\delta} \; ; \quad \epsilon_m \leq \sqrt{\frac{1}{2}\epsilon_\phi + \frac{d_s}{4}\log 2\pi},$$

*where $d_a$ and $d_s$ denotes the dimension of action and state space.*

*Proof.* We first consider $\epsilon_\pi$. The behavior cloning objective in its supremum form is,

$$\begin{aligned}
\epsilon_\beta &= \sup_{s \in \mathcal{D}} \mathbb{E}_{a \sim \mathcal{D}(-|s)}[\|a - \hat{\pi}_\beta(s)\|_2^2 / 2] - \mathcal{H}(\pi_b(-|s)) \\
&= \sup_{s \in \mathcal{D}} \mathbb{E}_{a \sim \mathcal{D}(-|s)} \left[ -\log \pi_{\theta_0}(a|s) \right] - \mathcal{H}(\pi_b(-|s)) - \frac{d_a}{2}\log 2\pi \\
&= \sup_{s \in \mathcal{D}} D_{KL}(\pi_b(-|s)||\pi_{\theta_0}(-|s)) - \frac{d_a}{2}\log 2\pi.
\end{aligned}$$

We apply Pinsker's inequality to the true and estimated behavior policy to yield

$$\sup_s D_{TV}(\pi_b(-|s)||\pi_{\theta_0}(-|s)) \leq \sqrt{\frac{1}{2}\epsilon_\beta + \frac{d_a}{4}\log 2\pi}.$$

By the same Pinsker's inequality, we have,

$$\sup_s D_{TV}(\pi_{\theta_k}(-|s)||\pi_{\theta_{k+1}}(-|s)) \leq \sqrt{\delta/2}.$$

Therefore, by triangle inequality, we have

$$\sup_s D_{TV}(\pi_b(-|s)||\pi_{\theta_T}(-|s)) \leq \sqrt{\frac{1}{2}\epsilon_\beta + \frac{d_a}{4}\log 2\pi} + T\sqrt{\frac{1}{2}\delta} = \epsilon_\pi,$$

as desired.

We perform similarly for $\epsilon_m$. The model dynamics loss is

$$
\begin{aligned}
\epsilon_\phi & = \sup_{s,a} \mathbb{E}_{s' \sim \mathcal{D}(-|s,a)} \left[ \|s' - \hat{f}_\phi(s,a)\|_2^2/2 \right] - \mathcal{H}(p(-|s,a)) \\
& = \sup_{s,a} \mathbb{E}_{s' \sim \mathcal{D}(-|s,a)} \left[ -\log p_\phi(s'|s,a) \right] - \mathcal{H}(p(-|s,a)) - \frac{d_s}{2} \log 2\pi \\
& = \sup_{s,a} D_{KL}(p(-|s,a)||p_\phi(-|s,a)) - \frac{d_s}{2} \log 2\pi.
\end{aligned}
$$

We apply Pinsker's inequality to the true dynamics and learned model to yield

$$
\epsilon_m \le \sup_{s,a} D_{TV}(p(-|s,a)||p_\phi(-|s,a)) \le \sqrt{\frac{1}{2}\epsilon_\phi + \frac{d_s}{4} \log 2\pi},
$$

as desired.

## B TRADE-OFF BETWEEN SAMPLE AND DEPLOYMENT EFFICIENCY

An important aspect of deployment efficiency is the trade-off between sample and deployment efficiency. To collect multiple data points per experiment and show this trade-off, we run recursive BREMEN with different batch sizes, and record how many samples are required to cross different reward thresholds.

HalfCheetah (Reward 7,000 result) and other results from Figure 5 generally show that high deployment efficiency lowers sample efficiency, confirming the inherent trade-off. However, in rare cases, e.g. Ant (Reward 5,000 result), it could be possible to achieve both high deployment efficiency and high sample efficiency through the right choice of the batch size hyper-parameter.

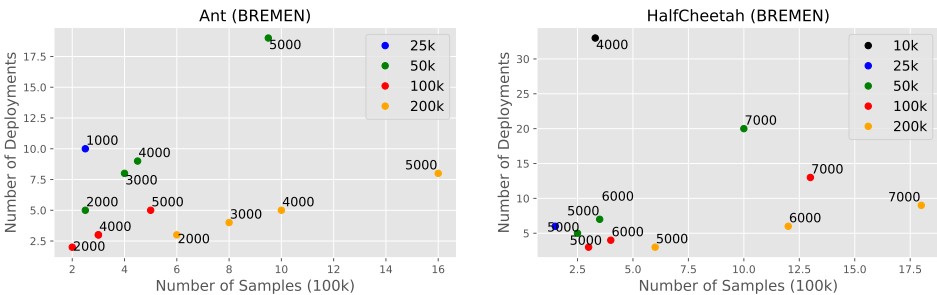

Figure 5: From the view of both sample and deployment efficiency at certain cumulative reward threshold, we evaluate BREMEN in Ant (left) and HalfCheetah (right). x and y axes respectively represent the number of samples and the number of deployments. Each data point comes from running BREMEN with different reward thresholds and batch sizes. The numbers above the points (e.g. 1000, 2000, ...) represent the reward threshold. The results (especially, reward 7,000 threshold in HalfCheetah) generally show that high deployment efficiency lowers sample efficiency, confirming the inherent trade-off.

## C    Discussion: Importance of Deployment Efficiency in Real-World Applications

Our notion of deployment-efficiency is necessitated by cost and safety constraints typical in many real world scenarios. Namely, a common approach to real-world applications (Cabi et al., 2020; Dulac-Arnold et al., 2019; Kalashnikov et al., 2018) is the following iterative training and data-collection paradigm:

1. Aggregate past previous dataset from worker(s)
2. Update policy based on the collected data
3. Deploy the policy to the worker(s)
4. Monitor the policy works as expected e.g. checking if it does not violate safety criterion (this safety verification step may alternatively happen before step 3)
5. Let the worker(s) collect experiences with the latest policy.

It is easy to see that the number of deployments is a critical bottleneck, as it involves both monitoring of the policy (Step 4) and communication to the workers (Step 3), and both of these steps can incur significant cost. Specifically, Step 4 requires evaluating the policy for safety, and often requires human monitors (Atkeson et al., 2015). As for Step 3, communication to workers can also be a bottleneck, especially in highly-parallelized distributed RL systems (Nair et al., 2015; Espeholt et al., 2018; 2019). Every policy deployment requires a potentially expensive communication between different machines/processes, and this can be a bottleneck on the whole system.

As a concrete example of the necessity of good deployment efficiency, consider optimization of personalization in web apps or recommender systems (Abel et al., 2017). Once a policy is learned on a batch of past experiences, it is deployed to a collection of web-hosting servers. In this scenario, both safety and communication concerns are relevant: Safety of the new policy is typically ensured by initially deploying the policy to a small percentage of users; after monitoring the results for some length of time (e.g. the newly deployed policy does not deteriorate user experiences), one can expand the target user set. As for communication, deploying a new policy to web-hosting servers can be time intensive, especially in large-scale web applications where the policy must be deployed to a network of servers around the world. Thus, in this setting, it is clear that online updating of the policy is infeasible due to both safety and communication constraints. Accordingly, the deployment-efficiency of any candidate RL algorithm is of tantamount importance.

The safe exploration might be mentioned as a potential alternative to deployment-efficiency. While safe exploration can arguably tackle the first concern above (safety risks of the policy), it does nothing to mitigate the latter (the engineering or communication costs associated with online deployment of a policy). Furthermore, this still ignores the fact that in many scenarios the ability to do safe exploration is not a given. While some safe RL algorithms can provide guarantees in tabular cases, these guarantees no longer hold when using function approximation with neural networks (Chow et al., 2018). In these cases, it can be much more difficult to perform "safe exploration" than it is to develop a deployment-efficient algorithm.

# D EVALUATING OFFLINE PERFORMANCES ON D4RL DATASETS

We compare BREMEN to MOPO (Yu et al., 2020), concurrently proposed model-based offline methods penalized by model epistemic uncertainty, and state-of-the-art model-free offline algorithms, namely, CQL (Kumar et al., 2020), BEAR (Kumar et al., 2019), BRAC (Wu et al., 2019), AWR (Peng et al., 2019) and BCQ (Fujimoto et al., 2019), on the D4RL MuJoCo locomotion datasets (Fu et al., 2020), used as standard offline RL benchmarks (Kumar et al., 2020; Nair et al., 2020). They have several types of offline data collected with different strategies. We choose the hyper parameters of BREMEN in Section 5.1 and F.2.2. Table 2 shows BREMEN beats recent state-of-the-art algorithms with the highest normalized score (around 100 corresponds to an expert) in several tasks, while none of the methods consistently achieves the best performance. This result suggests that the implicit regularization with the model-based method performs surprisingly well in offline settings despite of its simplicity.

| Task Name | BC | BREMEN | MOPO | CQL | BEAR | BRAC-v | AWR | BCQ |
|---|---|---|---|---|---|---|---|---|
| halfcheetah-random | 2.1 | **36.9** | 31.9 | 35.4 | 25.1 | 31.2 | 2.5 | 2.2 |
| walker2d-random | 1.6 | 3.7 | **13.0** | 7.0 | 7.3 | 1.9 | 1.5 | 4.9 |
| hopper-random | 9.8 | 12.2 | **13.3** | 10.8 | 11.4 | 12.2 | 10.2 | 10.6 |
| halfcheetah-medium | 36.1 | **55.0** | 40.2 | 44.4 | 41.7 | 46.3 | 37.4 | 40.7 |
| walker2d-medium | 6.6 | 59.6 | 14.0 | 79.2 | 59.1 | **81.1** | 17.4 | 53.1 |
| hopper-medium | 29.0 | **69.3** | 26.5 | 58.0 | 52.1 | 31.1 | 35.9 | 54.5 |
| halfcheetah-medium-replay | 38.4 | 47.2 | **54.0** | 46.2 | 38.6 | 47.7 | 40.3 | 38.2 |
| walker2d-medium-replay | 11.3 | 7.6 | **42.7** | 26.7 | 19.2 | 0.9 | 15.5 | 15.0 |
| hopper-medium-replay | 11.8 | 24.1 | **92.5** | 48.6 | 33.7 | 0.6 | 28.4 | 33.1 |
| halfcheetah-medium-expert | 35.8 | 53.3 | 57.9 | **62.4** | 53.4 | 41.9 | 52.7 | 64.7 |
| walker2d-medium-expert | 6.4 | 55.2 | 55.0 | **98.7** | 40.1 | 81.6 | 53.8 | 57.5 |
| hopper-medium-expert | **111.9** | 64.6 | 51.7 | 111.0 | 96.3 | 0.8 | 27.1 | 110.9 |

Table 2: Evaluation on D4RL MuJoCo locomotion datasets. The normalized score of BREMEN are averaged over 4 random seeds. We refer the score of MOPO (Yu et al., 2020) and CQL (Kumar et al., 2020) from their original papers. Other results are cited from Fu et al. (2020). BREMEN achieves the best and competitive score in several domains, while none of the algorithms beats all other methods.

# E INCORPORATING PESSIMISTIC MODEL-BASED OFFLINE METHODS INTO BREMEN

The concurrent model-based offline RL methods prescribe the use of uncertainty-based penalties (Kidambi et al., 2020; Yu et al., 2020), which can be incorporated into BREMEN. We therefore augmented BREMEN with either a *hard* (MOReL-like, green) or *soft* (MOPO-like, orange) reward penalty according to model uncertainty. MOReL quantifies the uncertainty measuring the maximum discrepancy of the prediction across the ensembles of the models and receives constant negative reward (-5.0 in our experiments) if the discrepancy is larger than the threshold (we set 3.0). MOPO measures the uncertainty by the maximum standard deviation of the model ensembles and uses this as a reward penalty with a coefficient (0.1 in our experiments). Evaluations in Figure 6 reveal that the soft reward penalty has notable results in Hopper and Walker2d, where model uncertainty is more crucial due to the episode's termination. Hard reward penalty seems overly pessimistic in deployment-efficient settings.

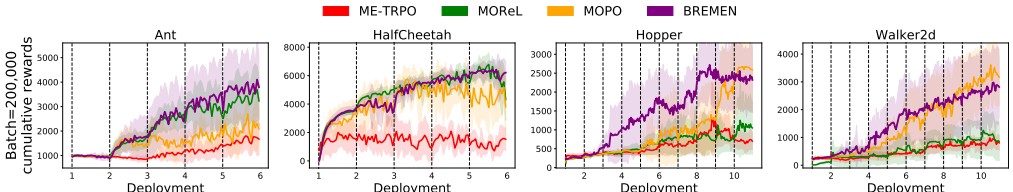

Figure 6: Comparison to the pessimistic reward shaping incorporated into BREMEN. Soft reward penalty (MOPO-like, orange) performs well in the environments where the incomplete models appear to be fatal.

# F    DETAILS OF EXPERIMENTAL SETTINGS

## F.1    IMPLEMENTATION DETAILS

For our baseline methods, we use the open-source implementations of SAC, BC, BCQ, and BRAC published in Wu et al. (2019). SAC and BRAC have (300, 300) Q-Network and (200, 200) policy network. BC has (200, 200) policy network, and BCQ has (300, 300) Q-Network, (300, 300) policy network, and (750, 750) conditional VAE. As for online ME-TRPO, we utilize the codebase of model-based RL benchmark (Wang et al., 2019). BREMEN and online ME-TRPO use the policy consisting of two hidden layers with 200 units. The dynamics model also consists of two hidden layers with 1,024 units. We use Adam (Kingma & Ba, 2014) as the optimizer with the learning rate of 0.001 for the dynamics model, and 0.0005 for behavior cloning in BREMEN. Especially in BREMEN and online ME-TRPO, we adopt a linear feature value function to stabilize the training. BREMEN in deployment-efficient settings takes about two or three hours per deployment on an NVIDIA TITAN V.

To leverage neural networks as Dyna-style (Sutton, 1991) dynamics models, we modify reward and termination function so that they are not dependent on the internal physics engine for calculation, following model-based benchmark codebase (Wang et al., 2019); see Table 3. Note that the score of baselines (e.g., BCQ, BRAC) is slightly different from Wu et al. (2019) due to this modification of the reward function. We re-run each algorithm in our environments and got appropriate convergence.

The maximum length of one episode is 1,000 steps without any termination in Ant and HalfCheetah; however, termination function is enabled in Hopper and Walker2d. The batch size of transitions for policy update is 50,000 in BREMEN and ME-TRPO, following Kurutach et al. (2018). The batch size of BC and BRAC is 256, and BCQ is 100, also following Wu et al. (2019).

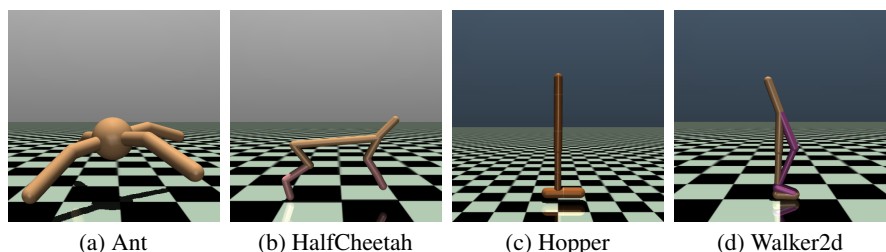

| (a) Ant | (b) HalfCheetah | (c) Hopper | (d) Walker2d |

Figure 7: Four standard MuJoCo benchmark environments used in our experiments.

| Environment | Reward function | Termination in rollouts |
|---|---|---|
| Ant | $\dot{x}_t - 0.1\|\boldsymbol{a}_t\|_2^2 - 3.0 \times (z_t - 0.57)^2 + 1$ | False |
| HalfCheetah | $\dot{x}_t - 0.1\|\boldsymbol{a}_t\|_2^2$ | False |
| Hopper | $\dot{x}_t - 0.001\|\boldsymbol{a}_t\|_2^2 + 1$ | True |
| Walker2d | $\dot{x}_t - 0.001\|\boldsymbol{a}_t\|_2^2 + 1$ | True |

Table 3: Reward function and termination in rollouts in the experiments. We remove all contact information from observation of Ant, basically following Wang et al. (2019).

## F.2 HYPER PARAMETERS

In this section, we describe the hyper-parameters in both deployment-efficient RL (Section F.2.1) and offline RL (Section F.2.2) settings. We run all of our experiments with five random seed, and the results are averaged.

### F.2.1 DEPLOYMENT-EFFICIENT RL

Table 4 shows the hyper-parameters of BREMEN. The rollout length is searched from {250, 500, 1000}, and max step size $\delta$ is searched from {0.001, 0.01, 0.05, 0.1, 1.0}. As for the discount factor $\gamma$ and GAE $\lambda$, we follow Wang et al. (2019).

| Parameter | Ant | HalfCheetah | Hopper | Walker2d |
|---|---|---|---|---|
| Iteration per batch | 2,000 | 2,000 | 6,000 | 2,000 |
| Deployment | 5 | 5 | 10 | 10 |
| Total iteration | 10,000 | 10,000 | 60,000 | 20,000 |
| Rollouts length | 250 | 250 | 1,000 | 1,000 |
| Max step size $\delta$ | 0.05 | 0.1 | 0.05 | 0.05 |
| Discount factor $\gamma$ | 0.99 | 0.99 | 0.99 | 0.99 |
| GAE $\lambda$ | 0.97 | 0.95 | 0.95 | 0.95 |
| Stationary noise $\sigma$ | 0.1 | 0.1 | 0.1 | 0.1 |

Table 4: Hyper-parameters of BREMEN in deployment-efficient settings.

**Number of Iterations for Policy Optimization**     To achieve high deployment efficiency, the number of iterations for policy optimization between deployments is one of the important hyper-parameters for fast convergence. In the existing methods (BCQ, BRAC, SAC), we search over three values: {10,000, 50,000, 100,000}, and choose 10,000 in BCQ and BRAC, and 100,000 in SAC (Figure 8). For BREMEN, we also search over three values: {2,000, 4,000, 6,000}. Figure 9 shows the results of iteration search, and we choose 2,000 in Ant, HalfCheetah, and Walker2d, and 6,000 in Hopper.

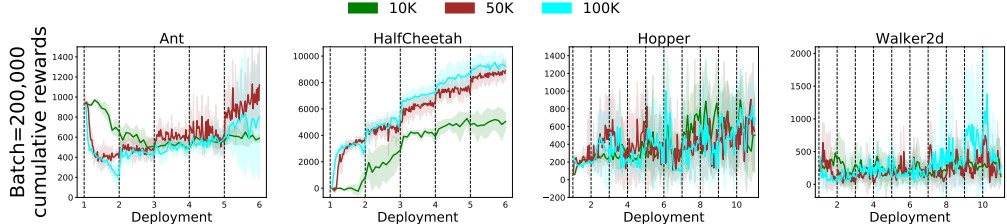

Figure 8: Search on the number of iterations for SAC policy optimization between deployments. The number of transitions per one data-collection is 200K.

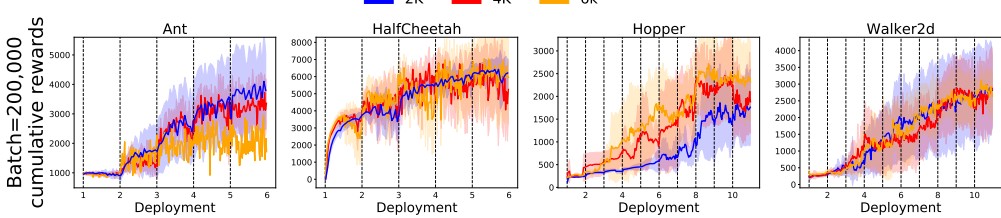

Figure 9: Search on the number of iterations for BREMEN policy optimization between deployments. The number of transitions per one data-collection is 200K.

**Stationary Noise in BREMEN**   To achieve effective exploration, the stochastic Gaussian policy is a good choice. We found that adding stationary Gaussian noise to the policy in the imaginary trajectories and data collection led to the notable improvement. Stationary Gaussian policy is written as,

$$a_t = \tanh(\mu_\theta(s_t)) + \epsilon, \qquad \epsilon \sim \mathcal{N}(0, \sigma^2).$$

Another choice is a learned Gaussian policy, which parameterizes not only $\mu_\theta$ but also $\sigma_\theta$. Learned gaussian policy is also written as,

$$a_t = \tanh(\mu_\theta(s_t)) + \sigma_\theta(s_t) \odot \epsilon, \qquad \epsilon \sim \mathcal{N}(0, \sigma^2).$$

We utilize the zero-mean Gaussian $\mathcal{N}(0, \sigma^2)$, and tune up $\sigma$ in Figure 10 with HalfCheetah, comparing stationary and learned strategies. From this experiment, we found that the stationary noise, the scale of 0.1, consistently performs well, and therefore we used it for all our experiments.

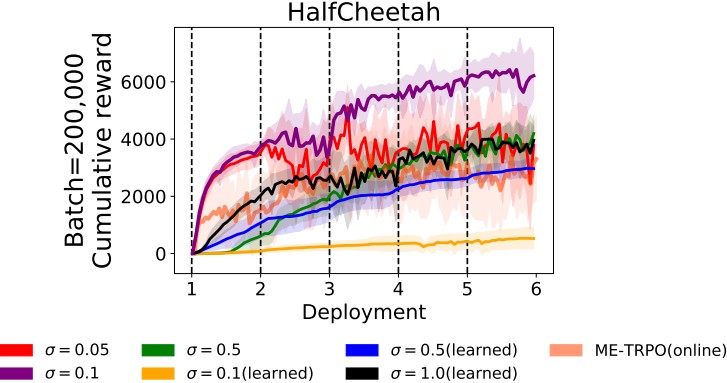

Figure 10: Search on the Gaussian noise parameter $\sigma$ in HalfCheetah. The number of transitions per one data-collection is 200K.

**Other Hyper-parameters in the Existing Methods**   As for online ME-TRPO, we collect 3,000 steps through online interaction with the environment per 25 iterations and split these transitions into a 2-to-1 ratio of training and validation dataset for learning dynamics models. In batch size 100,000 settings, we collect 2,000 steps and split with 1-to-1 ratio. Totally, we iterate 12,500 times policy optimization, which is equivalent to 500 deployments of the policy. Note that we carefully tune up the hyper-parameters of online ME-TRPO, and the performance is improved from Wang et al. (2019).

Table 5 and Table 6 shows the tunable hyper-parameters of BCQ and BRAC, respectively. We refer Wu et al. (2019) to choose these values. In this work, BRAC applies a primal form of KL value penalty, and BRAC (max Q) means sampling multiple actions and taking the maximum according to the learned Q function.

| Parameter | Ant | HalfCheetah | Hopper | Walker2d |
|---|---|---|---|---|
| Policy learning rate | 3e-05 | 3e-04 | 3e-06 | 3e-05 |
| Perturbation range $\Phi$ | 0.15 | 0.5 | 0.15 | 0.15 |

Table 5: Hyper-parameters of BCQ.

| Parameter | Ant | HalfCheetah | Hopper | Walker2d |
|---|---|---|---|---|
| Policy learning rate | 1e-4 | 1e-3 | 3e-5 | 1e-5 |
| Divergence penalty $\alpha$ | 0.3 | 0.1 | 0.3 | 0.3 |

Table 6: Hyper-parameters of BRAC.

### F.2.2 OFFLINE RL

In the offline experiments, we apply the same hyper-parameters as in the deployment-efficient settings described above, except for the iteration per batch. Algorithm 2 is pseudocode for BREMEN in offline RL settings where policies are updated only with one fixed batch dataset. The number of iteration $T$ is set to 6,250 in BREMEN, and 500,000 in BC, BCQ, and BRAC.

The datasets for 50k or 100k experiments are sliced from the beginning of the 1M batched datasets without shuffling, but we observed that the distribution of rewards in 50k or 100k is not different from 1M.

---

**Algorithm 2** BREMEN for Offline RL

---

**Input:** Offline dataset $\mathcal{D} = \{s_t, a_t, r_t, s_{t+1}\}$, Initial parameters $\phi = \{\phi_1, \cdots, \phi_K\}$, $\beta$, Number of policy optimization $T$.
1: Train $K$ dynamics models $\hat{f}_\phi$ using $\mathcal{D}$ via Equation 1.
2: Train estimated behavior policy $\hat{\pi}_\beta$ using $\mathcal{D}$ by behavior cloning via Equation 3.
3: Initialize target policy $\pi_{\theta_0} = \text{Normal}(\hat{\pi}_\beta, 1)$.
4: **for** policy optimization $k = 1, \cdots, T$ **do**
5:     Generate imaginary rollout.
6:     Optimize target policy $\pi_\theta$ satisfying Equation 4 with the rollout.

---

# G ADDITIONAL EXPERIMENTAL RESULTS

## G.1 PERFORMANCE ON THE DATASET WITH DIFFERENT NOISE

Following Wu et al. (2019) and Kidambi et al. (2020), we additionally compare BREMEN in offline settings to the other baselines (BC, BCQ, BRAC) with five datasets of different exploration noise. Each dataset has also one million transitions.

- **eps1**: 40 % of the dataset is collected by data-collection policy (partially trained SAC policy) $\pi_b$, 40 % of the dataset is collected by epsilon greedy policy with $\epsilon = 0.1$ to take a random action, and 20 % of dataset is collected by an uniformly random policy.

- **eps3**: Same as eps1, 40 % of the dataset is collected by $\pi_b$, 40 % is collected by epsilon greedy policy with $\epsilon = 0.3$, and 20 % is collected by an uniformly random policy.

- **gaussian1**: 40 % of the dataset is collected by data-collection policy $\pi_b$, 40 % is collected by the policy with adding zero-mean Gaussian noise $\mathcal{N}(0, 0.1^2)$ to each action sampled from $\pi_b$, and 20 % is collected by an uniformly random policy.

- **gaussian3**: 40 % of the dataset is collected by data-collection policy $\pi_b$, 40 % is collected by the policy with zero-mean Gaussian noise $\mathcal{N}(0, 0.3^2)$, and 20 % is collected by an uniformly random policy.

- **random**: All of the dataset is collected by an uniformly random policy.

Table 7 shows that BREMEN can also achieve performance competitive with state-of-the-art model-free offline RL algorithm even with noisy datasets. The training curves of each experiment are shown in Appendix G.4.

## G.2 COMPARISON AMONG DIFFERENT NUMBER OF ENSEMBLES

To deal with the distribution shift during policy optimization, also known as model bias, we introduce the dynamics model ensembles. We validate the performance of BREMEN with a different number of dynamics models $K$. Figure 11 and Figure 12 show the performance of BREMEN with the different number of ensembles in deployment-efficient and offline settings. Ensembles with more dynamics models resulted in better performance due to the mitigation of distributional shift except for $K = 10$, and then we choose $K = 5$.

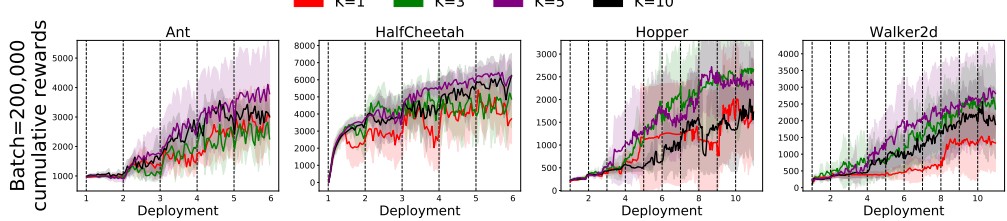

Figure 11: Comparison of the number of dynamics models in deployment-efficient settings.

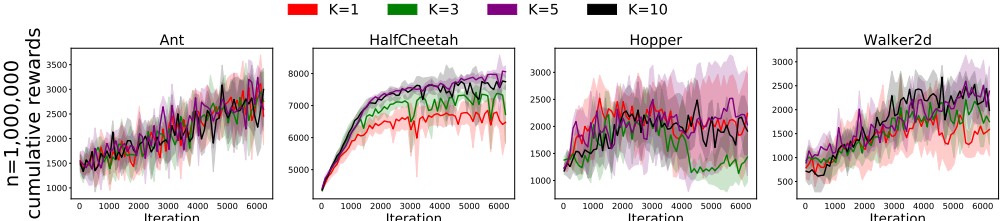

Figure 12: Comparison of the number of dynamics models in offline settings.

| Noise: eps1, 1,000,000 (1M) transitions | | | | |
|---|---|---|---|---|
| **Method** | **Ant** | **HalfCheetah** | **Hopper** | **Walker2d** |
| Dataset | 1077 | 2936 | 791 | 815 |
| BC | 1381±71 | 3788±740 | 266±486 | 1185±155 |
| BCQ | 1937±116 | 6046±276 | 800±659 | 479±537 |
| BRAC | 2693±155 | 7003±118 | 1243±162 | 3204±103 |
| BRAC (max Q) | 2907±98 | 7070±81 | 1488±386 | **3330±147** |
| BREMEN (Ours) | **3519±129** | **7585±425** | **2818±76** | 1710±429 |
| ME-TRPO (offline) | 1514±503 | 1009±731 | 1301±654 | 128±153 |
| **Noise: eps3, 1,000,000 (1M) transitions** | | | | |
| **Method** | **Ant** | **HalfCheetah** | **Hopper** | **Walker2d** |
| Dataset | 936 | 2408 | 662 | 648 |
| BC | 1364±121 | 2877±797 | 519±532 | 1066±176 |
| BCQ | 1938±21 | 5739±188 | 1170±446 | 1018±1231 |
| BRAC | 2718±90 | 6434±147 | 1224±71 | 2921±101 |
| BRAC (max Q) | 2913±87 | 6672±136 | 2103±746 | **3079±110** |
| BREMEN (Ours) | **3409±218** | **7632±104** | **2803±65** | 1586±139 |
| ME-TRPO (offline) | 1843±674 | 5504±67 | 1308±756 | 354±329 |
| **Noise: gaussian1, 1,000,000 (1M) transitions** | | | | |
| **Method** | **Ant** | **HalfCheetah** | **Hopper** | **Walker2d** |
| Dataset | 1072 | 3150 | 882 | 1070 |
| BC | 1279±80 | 4142±189 | 31±16 | 1137±477 |
| BCQ | 1958±76 | 5854±498 | 475±416 | 608±416 |
| BRAC | 2905±81 | 7026±168 | 1456±161 | 3030±103 |
| BRAC (max Q) | 2910±157 | 7026±168 | 1575±89 | **3242±97** |
| BREMEN (Ours) | **2912±165** | **7928±313** | **1999±617** | 1402±290 |
| ME-TRPO (offline) | 1275±656 | 1275±656 | 909±631 | 171±119 |
| **Noise: gaussian3, 1,000,000 (1M) transitions** | | | | |
| **Method** | **Ant** | **HalfCheetah** | **Hopper** | **Walker2d** |
| Dataset | 1058 | 2872 | 781 | 981 |
| BC | 1300±34 | 4190±69 | 611±467 | 1217±361 |
| BCQ | 1982±97 | 5781±543 | 1137±582 | 258±286 |
| BRAC | 3084±180 | 3933±2740 | 1432±499 | 3253±118 |
| BRAC (max Q) | 2916±99 | 3997±2761 | 1417±267 | **3372±153** |
| BREMEN (Ours) | **3432±185** | **8124±145** | **1867±354** | 2299±474 |
| ME-TRPO (offline) | 1237±310 | 2141±872 | 973±243 | 219±145 |
| **Noise: random, 1,000,000 (1M) transitions** | | | | |
| **Method** | **Ant** | **HalfCheetah** | **Hopper** | **Walker2d** |
| Dataset | 470 | -285 | 34 | 2 |
| BC | 989±10 | -2±1 | 106±62 | 108±110 |
| BCQ | 1222±114 | 2887±242 | 206±7 | 228±12 |
| BRAC | 1057±92 | 3449±259 | 227±30 | 29±54 |
| BRAC (max Q) | 683±57 | 3418±171 | 224±37 | 26±50 |
| BREMEN (Ours) | 905±11 | **3627±193** | 270±68 | 254±6 |
| ME-TRPO (offline) | **2221±665** | 2701±120 | **321±29** | **262±13** |

Table 7: Comparison of BREMEN to the existing offline methods in offline settings, namely, BC, BCQ (Fujimoto et al., 2019), and BRAC (Wu et al., 2019). Each cell shows the average cumulative reward and their standard deviation with 5 seeds. The maximum steps per episode is 1,000. Five different types of exploration noise are introduced during the data collection, eps1, eps3, gaussian1, gaussian3, and random. BRAC applies a primal form of KL value penalty, and BRAC (max Q) means sampling multiple actions and taking the maximum according to the learned Q function.

### G.3 IMPLICIT KL CONTROL IN OFFLINE SETTINGS

Similar to Section 5.3, we present offline RL experiments to better understand the effect of implicit KL regularization. In contrast to the implicit KL regularization with Equation 4, the optimization of BREMEN with explicit KL value penalty becomes

$$\theta_{k+1} = \arg\max_{\theta} \mathop{\mathbb{E}}_{s,a\sim\pi_{\theta_k},\hat{f}_{\phi_i}} \left[ \frac{\pi_\theta(a|s)}{\pi_{\theta_k}(a|s)} \left( A^{\pi_{\theta_k}}(s,a) - \alpha D_{\mathrm{KL}}(\pi_\theta(\cdot|s)\|\hat{\pi}_\beta(\cdot|s)) \right) \right] \qquad (6)$$
$$\text{s.t.} \quad \mathop{\mathbb{E}}_{s\sim\pi_{\theta_k}} \left[ D_{\mathrm{KL}}\left(\pi_\theta(\cdot|s)\|\pi_{\theta_k}(\cdot|s)\right) \right] \le \delta,$$

where $A^{\pi_{\theta_k}}(s,a)$ is the advantage of $\pi_{\theta_k}$ computed using imaginary rollouts with the learned dynamics model and $\delta$ is the maximum step size. Note that BREMEN with explicit KL penalty does not utilize behavior cloning initialization.

We empirically conclude that the explicit constraint $-\alpha D_{\mathrm{KL}}(\pi_\theta(\cdot|s)\|\hat{\pi}_\beta(\cdot|s))$ is unnecessary and just TRPO update with behavior-initialization as implicit regularization is sufficient in BREMEN algorithm. Figure 13 shows the KL divergence between learned policies and the last deployed policies (top row) and model errors measured by a mean squared error of predicted next state from the true state (second row). We find that behavior initialized policy with conservative KL trust-region updates well stuck to the last deployed policy during improvement without explicit KL penalty. The policy initialized with behavior cloning also tended to suppress the increase of model error, which implies that behavior initialization alleviates the effect of the distribution shift. In Walker2d, the model error of BREMEN is relatively large, which may relate to the poor performance with noisy datasets in Section G.1.

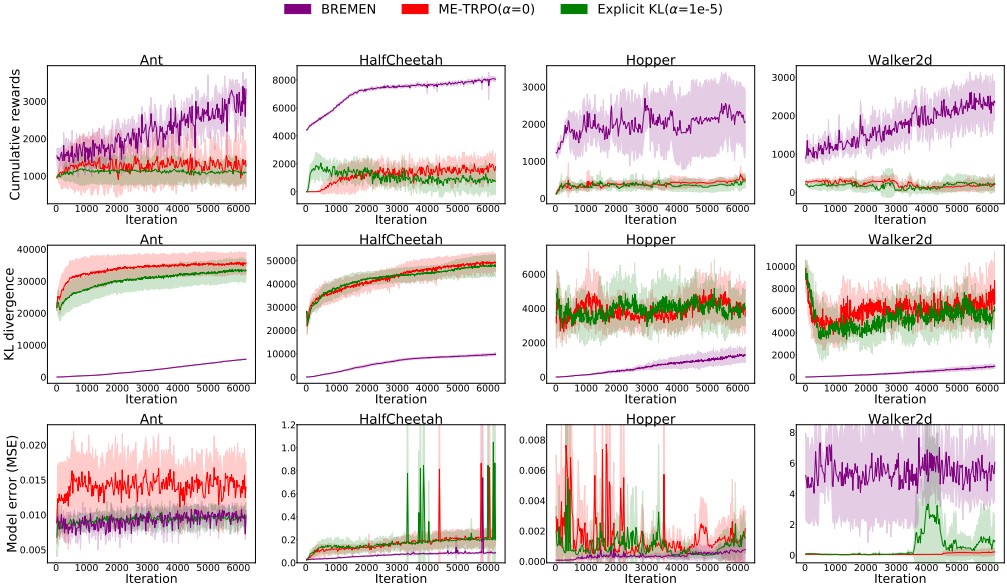

Figure 13: Average cumulative rewards (top row) and corresponding KL divergence of learned policies from the last deployed policy (second row) and model errors (bottom row) in offline settings with 1M dataset (no noise). Behavior initialized policy (purple line) tends to suppress the policy and model error during training better than no-initialization (red line) or explicit KL penalty (green line).

## G.4 TRAINING CURVES FOR OFFLINE RL WITH DIFFERENT NOISES

In this section, we present training curves of our all experiments in offline settings. Figure 14 shows the results in Section 5.1. Figure 15, 16, 17, 18, and 19 also show the results in Section G.1.

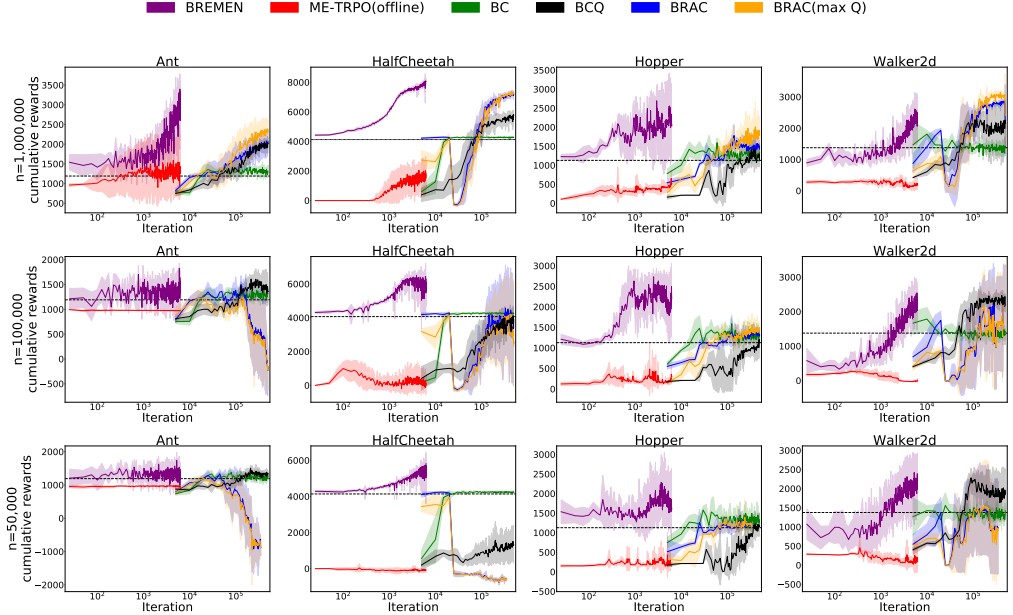

Figure 14: Performance in Offline RL experiments (Table 1). (top row) dataset size is 1M, (second row) 100K, and (bottom row) 50K, respectively. Note that x-axis is the number of iterations with policy optimization in a log-scale.

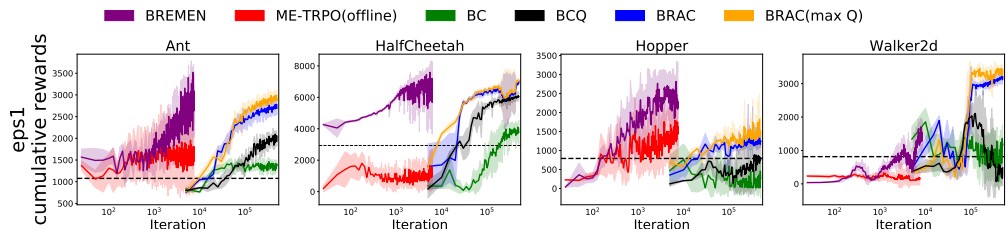

Figure 15: Performance in Offline RL experiments with $\epsilon$-greedy dataset noise $\epsilon = 0.1$. Dataset size is 1M.

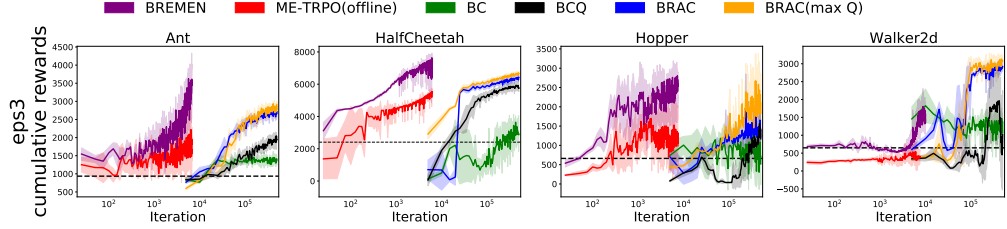

Figure 16: Performance in Offline RL experiments with $\epsilon$-greedy dataset noise $\epsilon = 0.3$. Dataset size is 1M.

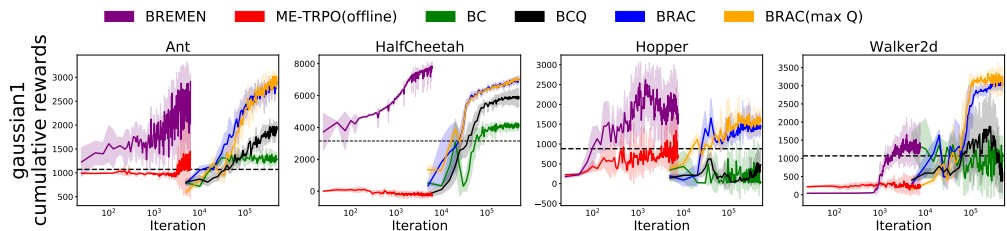

Figure 17: Performance in Offline RL experiments with gaussian dataset noise $\mathcal{N}(0, 0.1^2)$. Dataset size is 1M.

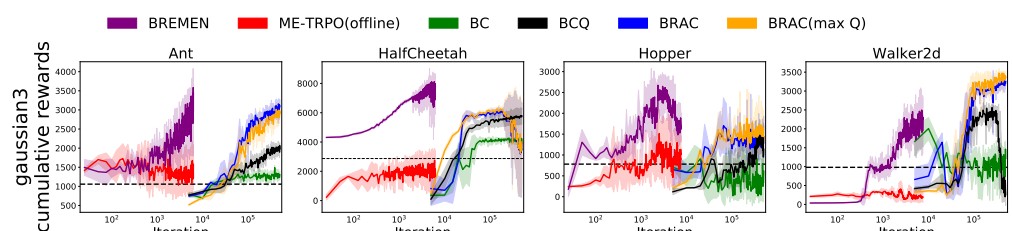

Figure 18: Performance in Offline RL experiments with gaussian dataset noise $\mathcal{N}(0, 0.3^2)$. Dataset size is 1M.

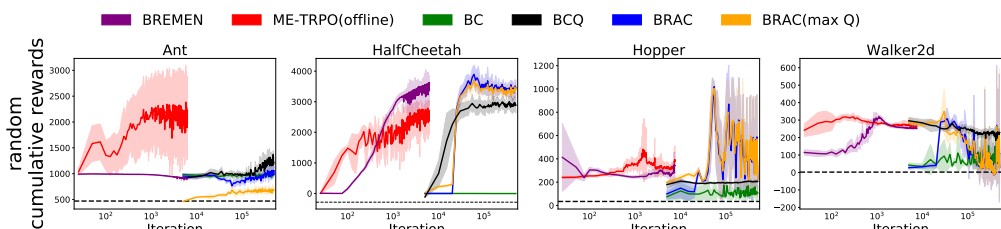

Figure 19: Performance in Offline RL experiments with completely random behaviors. Dataset size is 1M.

### G.5    DEPLOYMENT-EFFICIENT RL EXPERIMENT WITH DIFFERENT REWARD FUNCTION

In addition to the main results in Section 5.2 (Figure 2), we also evaluate BREMEN in deployment-efficient setting with different reward function. We modified HalfCheetah environment into the one similar to cheetah-run task in Deep Mind Control Suite.[5] The reward function is defined as

$$r_t = \begin{cases} 0.1\dot{x}_t & (0 \le \dot{x}_t \le 10) \\ 1 & (\dot{x}_t > 10), \end{cases}$$

and the termination is turned off. Figure 20 shows the performance of BREMEN and existing methods. BREMEN also shows better deployment efficiency than other existing offline methods and online ME-TRPO, except for SAC, which is the same trend as that of main results.

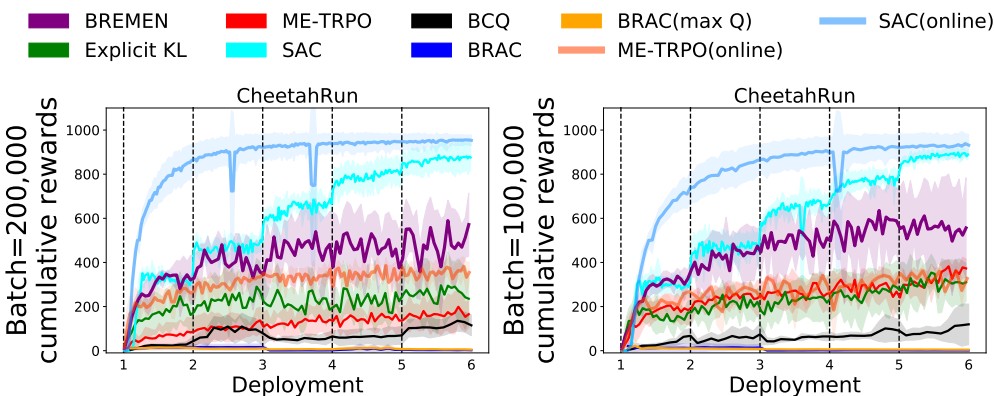

Figure 20: Performance in Deployment-Efficient RL experiments with different reward function of HalfCheetah.

---

[5]https://github.com/deepmind/dm_control/blob/master/dm_control/suite/cheetah.py

