# OpenReview forum: "Deployment-Efficient Reinforcement Learning via Model-Based Offline Optimization"
_ICLR.cc/2021/Conference — ICLR 2021 Poster_

### Official Review · AnonReviewer3 · 2020-10-24
**simple yet effective method for achieving deployment-efficiency (but requiring some prior knowledge?)**

**Rating:** 8
**Confidence:** 4

**Review:**

The paper explores an under-researched problem, that of minimizing the number of policy updates in an RL setting (or being “deployment efficient”). This is an important aspect of using RL agents in real “production” environments where there may be many reasons why updates are costly and limiting them is an important consideration in the choice of RL method (or whether to even use RL).

The paper shows that so-called "off-policy” methods which, by their naming as such, it is implied that they should work in a sparse-deployments environment are, in fact, not suited (and often not evaluated) for this regime.

By introducing a set of simple and strait-forward steps to the update and deployment process, the paper shows performance that approaches the continuous-deployment performance of comparable un-constrained methods.

The main ingredients of the proposed method seem to be:
Model based approach to support model-based offline exploration (an ensemble of models to prevent exploitation of model inaccuracies)
Re-estimation of a model by Behaviour Cloning (BC) with data from last deployment (appears to have a large contribution though I don’t understand why)
Conservative off-line policy updates (constrained by KL to BC policy) using offline rollouts (with forward model ensemble)

The evaluation presented in the paper and extensive (13-page) appendix clearly show the advantage of the proposed method in the sparse-deployments regime and the overall competitive performance with regards to sample efficiency as well. Code was made available as supplementary material.

I am unclear on why each training iteration should start with a policy learned from behavioural cloning of the last deployment’s data instead of the model that was deployed which would be available at that time. Figure 4 clearly shows BC to be the better approach In practice but I would appreciate some intuitive reasoning for this (unless this is standard practice).

Perhaps my main concern with this paper, given the problem it addresses - that of a real deployment setting, is that it seems that some of the parameters that need to be tuned to achieve great performance require fitting to the specific task (i.e. deployments !). As far as I understand it, parameters such as the number of offline policy updates per deployment or the weight of the KL-divergence in the policy update step are crucial to good performance yet the paper does not explain how to choose them without engaging in the true environment. If that is correct than this seems to defeat the aims of the paper and question the overall methodology for deployment-efficiency.

---

> ### Author Response · Authors · 2020-11-17
> **Author Response to AnonReviewer3**
>
> We appreciate the reviewer for the helpful feedback.
>
> **> “why each training iteration should start with a policy learned from behavioral cloning of the last deployment’s data”**
> We empirically verified the need for its re-initialization in Section 5.3, Figure 4. One possible explanation is that since each batch only contains a few finite samples from the policy action distribution per state, refitting the distribution to match the empirical sample distribution makes the policy optimization more precisely conservative around the “observed action data”, the premise of offline RL methods.
>
> **> Parameter tuning**
> Many real-world end-to-end RL papers [1, 2, 3] tune generic hyperparameters first in simulation per task or with similar tasks, and then use those for learning in the real world. Their results indicate that even a crude simulation model for each real-world task is generally sufficient to find those sufficiently-good hyper-parameters. We believe the same should apply to our approach: once one suitably tunes hyperparameters to derive a deployment-efficient RL algorithm in simulation, the algorithm & hyperparameters can then be applied to the real-world application, learning the real-world task in a minimal number of deployments.
> In addition, while we included the best results in the paper, the best hyper-parameters we used don’t vary too significantly per environment (See Appendix F.2.1). BREMEN’s robustness over hyper-parameters can also avoid the need for exhaustive hyperparameter search in real-world tasks.
>
> ```
> [1] Shixiang Gu, Ethan Holly, Timothy Lillicrap, and Sergey Levine. Deep Reinforcement Learning for Robotic Manipulation with Asynchronous Off-Policy Updates. In International Conference on Robotics and Automation, 2017.
> [2] Dmitry Kalashnikov, Alex Irpan, Peter Pastor, Julian Ibarz, Alexander Herzog, Eric Jang, Deirdre Quillen, Ethan Holly, Mrinal Kalakrishnan, Vincent Vanhoucke, and Sergey Levine. QT-Opt: Scalable Deep Reinforcement Learning for Vision-Based Robotic Manipulation. In Conference on Robot Learning, 2018.
> [3] Haarnoja, Tuomas, et al. "Soft actor-critic algorithms and applications." arXiv preprint arXiv:1812.05905 (2018).
> ```

---

### Official Review · AnonReviewer1 · 2020-10-28

**Rating:** 7
**Confidence:** 3

**Review:**

Summary

This paper proposes a new approach to learning control policies with improved data efficiency and fewer number of data collection sessions (with each session using a different policy). Further, the authors proposed a new concept of “deployment efficiency”, with a new “deployment” referring to using a new policy to interact with the real environment, for example, for data collection. The new approach belongs to the family of model-based online reinforcement learning algorithms and seems to primarily augment a prior approach, called ME-TRPO, by using a helper policy trained by behavior cloning data collected after the most recent deployment of learner policy. The experiment results validate that the proposed approach achieves better data efficiency and deployment efficiency compared to prior approaches.

==============

Positives

The experiments are very extensive and of high quality. A diverse set of tasks is used, along with a good range of prior approaches for comparison. The experiments are also well designed,   with one focusing on data efficiency in offline setting and the other one mimicking practical usage in online setting with limited number of deployments.

The experiment results are clear and convincing. Performance is the best in almost all tasks.

It is a great step towards wider application of RL algorithms in the real-world by recognizing practically important metric such as deployment efficiency.

The authors did a good job in introducing the varied set of related works and the general background, providing a clear organization and classification of prior approaches.

==============

Negatives

The authors could make a clear statement on the contributions of the paper. Several features of the proposed algorithm can be found in the prior approach of ME-TRPO, including all steps shown in Algorithm 1 except steps 2 & 5 & 6, i.e.: the use of an ensemble of learnt dynamics models to mitigate model bias and training instability, the use of imaginary rollout, the choice of TRPO for policy optimization, all are present in ME-TRPO. The clear innovation in this paper seems to be in using the behavior policy to bootstrap policy learning in a new iteration of deployment. The authors could make clear the approach is an extension of ME-TRPO, which at present is not in the way the authors proposed the algorithm.

While I don’t dispute the merit of deployment efficiency, it can be better motivated. The first paragraph of Section 3 mentions “verify each policy before deployment”; it’d be good to show references of real world applications of this procedure. It’s conceivable in practical settings, there can be better ways to achieve the verification in a way that circumvents the cost of deployment (for example, limiting policy outputs to a safe range at runtime, instead of “checking out-of-bounds actions” as a way of pre-validation).

==============

Recommendation

Overall the proposed approach clearly yields empirical advantages of prior approaches in both data efficiency and the newly proposed metric of deployment efficiency. The paper suffers from lack of algorithmic novelty, but the experimental efficiency outweighs.

---

> ### Author Response · Authors · 2020-11-17
> **Author Response to AnonReviewer1**
>
> We thank the reviewer for the careful reading of the paper.
>
> **> ”The authors could make a clear statement on the contributions of the paper.”**
> Thank you for the valuable comments! We will edit the manuscript to make our contributions clearer. The key contributions of our paper are (1) the proposal/evaluation of deployment-efficiency metric, (2) the proposal/evaluation of sample-efficient offline RL benchmarks (20x less data), and (3) the development of a simple algorithm for (1)+(2) through a novel combination of existing ideas (model-ensemble+trust-region from ME-TRPO, and BC initialization from offline literatures).
>
> **> References of real-world applications**
> Thank you for this suggestion! We will add additional references for real-world applications. For example, in robotic control literature, Atkeson et al. [1] introduces safety features to detect potential catastrophic failures and stop its behavior, but it requires a human supervisor, which is highly costly.
>
> ~We will upload the edited draft in the next few days.~ *We uploaded the revised version on 24th Nov.*
>
> ```
> [1] C. G. Atkeson, B. P.W. Babu, N. Banerjee, D. Berenson, C. P. Bove, X. Cui, M. DeDonato, R. Du, S. Feng, P. Franklin, M. Gennert, J. P. Graff, P. He, A. Jaeger, J. Kim, K. Knoedler, L. Li, C. Liu, X. Long, T. Padir, F. Polido, G. G. Tighe, and X. Xinjilefu. No falls, no resets: Reliable humanoid behavior in the DARPA robotics challenge. In International Conference on Humanoid Robots, 2015.
> ```

---

### Official Review · AnonReviewer4 · 2020-10-30
**Justification of Deployment Efficiency metric is unclear**

**Rating:** 5
**Confidence:** 4

**Review:**

The paper has been well written and is easy to understand. The premise of the paper is clear, and methods have been evaluated properly.

Suggestions to improve the paper:
- While the deployment efficiency metric has been clearly explained, the justification of this metric has not been elaborated on sufficiently. For the community to adopt a new metric, the paper needs to provide enough evidence to show that deploying policy is expensive, time consuming or requires manual curation. What are the steps involved in deployment? In what situations is deployment expensive? How common are they? Sample efficiency is a well established metric because it makes sense that we need to learn a good policy in as few real world interactions as possible. One can argue that safe exploration is essential for real world deployment. However, that is neither discussed nor compared with.
- The lack of clarity in the deployment efficiency metric also appears in the evaluation. Why does a batch size of 100K make sense? Why limit it to 5 to 10 deployments? These numbers appear to be arbitrary.
- How does one decide the trade off between sample efficiency and deployment efficiency?
-  A clarification question: How is the 50k and 100k transition obtained for evaluating sample efficiency? Is it sampled randomly from the 1M dataset or is it sliced without shuffling?
- The BREMEN algorithm borrows ideas from related prior works, and is reasonable. However, it is unclear why the algorithm is better suited to improve deployment efficiency compared to prior works. From the explanation provided in the paper, BREMEN is a good offline RL algorithm and is coincidentally better at deployment efficiency as well.

---

> ### Author Response · Authors · 2020-11-17
> **Author Response to AnonReviewer4 (1/2)**
>
> We thank the reviewer for the careful reading of the paper. We address concerns raised by the reviewer below.
>
> **> Justification of deployment efficiency metric**
> Our notion of deployment-efficiency is necessitated by cost and safety constraints typical in many real-world scenarios. Namely, a common approach to real-world applications [1, 2, 3] is the following iterative training and data-collection paradigm:
> 1. Aggregate past previous dataset from worker(s)
> 2. Update policy based on the collected data
> 3. Deploy the policy to the worker(s)
> 4. Monitor the policy works as expected e.g. checking if it does not violate safety criterion (this safety verification step may alternatively happen before step 3
> 5. Let the worker(s) collect experiences with the latest policy.
>
> It is easy to see that the number of deployments is a critical bottleneck, as it involves both monitoring of the policy (Step 4) and communication to the workers (Step 3), and both of these steps can incur significant cost. Specifically, Step 4 requires evaluating the policy for safety, and often requires human monitors [4]. As for Step 3, communication to workers can also be a bottleneck, especially in highly-parallelized distributed RL systems [5, 6, 7]. Every policy deployment requires a potentially expensive communication between different machines/processes, and this can be a bottleneck on the whole system.
>
> *[Edited this paragraph on 24th Nov.]* As a concrete example of the necessity of good deployment efficiency, consider optimization of personalization in web apps or recommender systems [8]. Once a policy is learned on a batch of past experiences, it is deployed to a collection of web-hosting servers. In this scenario, both safety and communication concerns are relevant:
> - Safety of the new policy is typically ensured by initially deploying the policy to a small percentage of users; after monitoring the results for some length of time (e.g. the newly deployed policy does not deteriorate user experiences), one can expand the target user set.
> - As for communication, deploying a new policy to web-hosting servers can be time-intensive, especially in large-scale web applications where the policy must be deployed to a network of servers around the world.
> Thus, in this setting, it is clear that online updating of the policy is infeasible due to both safety and communication constraints. Accordingly, the deployment-efficiency of any candidate RL algorithm is of tantamount importance.
>
> The reviewer mentions safe exploration as a potential alternative to deployment-efficiency. While safe exploration can arguably tackle the first concern above (safety risks of the policy), it does nothing to mitigate the latter (the engineering or communication costs associated with online deployment of a policy). Furthermore, this still ignores the fact that in many scenarios the ability to do safe exploration is not a given. While some safe RL algorithms can provide guarantees in tabular cases, these guarantees no longer hold when using function approximation with neural networks [9]. In these cases, it can be much more difficult to perform “safe exploration” than it is to develop a deployment-efficient algorithm.
>
> **> Justification of numbers in the experiments of deployment-efficiency**
> We agree that the numbers of deployment and batch size are somewhat arbitrary; however, since the batch size is typically set to 1M in standard offline RL experiments, we think our 5/10 deployments x 100/200 batch sizes are reasonable choices that allow fair comparisons with prior sample-efficient RL results while giving us the possibility to improve and evaluate deployment efficiency.
>
> **> Trade-off of sample and deployment efficiency** ***[Added on 24th Nov.]***
> As for the trade-off between sample and deployment efficiency, we added plots in Appendix B, where x and y axes respectively represent #samples and #deployments to reach certain reward thresholds. Each data point comes from running BREMEN with different batch sizes.  The results (especially, reward 7,000 threshold in HalfCheetah) generally show that high deployment efficiency lowers sample efficiency, confirming the inherent trade-off. We hope these additional results will help illuminate the deployment vs. sample efficiency trade-off.
>
>
> **> “How is the 50k and 100k transition obtained for evaluating sample efficiency?”**
> The transitions are sliced from the beginning of the 1M batched dataset without shuffling, but we observed that the distribution of rewards in 50k or 100k is not different from 1M. We added this clarification to experimental details in Appendix F.2.2. *(edited on 24th Nov.)*.

---

> > ### Author Response · Authors · 2020-11-17
> > **Author Response to AnonReviewer4 (2/2)**
> >
> > **> “it is unclear why the algorithm is better suited to improve deployment efficiency compared to prior works”**
> > We show in Section 5.1 that BREMEN is the most sample-efficient offline RL algorithm (works even with 20x less data than in prior benchmarks), and we regard this feature as directly contributing to a functional, sample-efficient algorithm in the limited-deployment settings. The inabilities of BCQ and BRAC to learn in small-data regimes make them unsuited for deployment-efficient and sample-efficient learning.
> >
> > ~We will upload the edited draft in the next few days.~ *We uploaded the revised version on 24th Nov.*
> >
> >
> > ```
> > [1] Serkan Cabi, Sergio Gómez Colmenarejo, Alexander Novikov, Ksenia Konyushkova, Scott Reed, Rae Jeong, Konrad Zolna, Yusuf Aytar, David Budden, Mel Vecerik, Oleg Sushkov, David Barker, Jonathan Scholz, Misha Denil, Nando de Freitas, and Ziyu Wang. Scaling data-driven robotics with reward sketching and batch reinforcement learning. arXiv preprint arXiv:1909.12200, 2019.
> > [2] Gabriel Dulac-Arnold, Daniel Mankowitz, and Todd Hester. Challenges of Real-World Reinforcement Learning. In International Conference on Machine Learning, 2019.
> > [3] Dmitry Kalashnikov, Alex Irpan, Peter Pastor, Julian Ibarz, Alexander Herzog, Eric Jang, Deirdre Quillen, Ethan Holly, Mrinal Kalakrishnan, Vincent Vanhoucke, and Sergey Levine. QT-Opt: Scalable deep reinforcement learning for vision-based robotic manipulation. In Conference on Robot Learning, 2018.
> > [4] C. G. Atkeson, B. P.W. Babu, N. Banerjee, D. Berenson, C. P. Bove, X. Cui, M. DeDonato, R. Du, S. Feng, P. Franklin, M. Gennert, J. P. Graff, P. He, A. Jaeger, J. Kim, K. Knoedler, L. Li, C. Liu, X. Long, T. Padir, F. Polido, G. G. Tighe, and X. Xinjilefu. No falls, no resets: Reliable humanoid behavior in the DARPA robotics challenge. In International Conference on Humanoid Robots, 2015.
> > [5] Arun Nair, Praveen Srinivasan, Sam Blackwell, Cagdas Alcicek, Rory Fearon, Alessandro De Maria, Vedavyas Panneershelvam, Mustafa Suleyman, Charles Beattie, Stig Petersen, et al. Massively parallel methods for deep reinforcement learning. arXiv preprint arXiv:1507.04296, 2015.
> > [6] Lasse Espeholt, Hubert Soyer, Remi Munos, Karen Simonyan, Vlad Mnih, Tom Ward, Yotam Doron, Vlad Firoiu, Tim Harley, Iain Dunning, et al. IMPALA: Scalable distributed deep-rl with importance weighted actor-learner architectures. In International Conference on Machine Learning, 2018.
> > [7] Lasse Espeholt, Raphaël Marinier, Piotr Stanczyk, Ke Wang, and Marcin Michalski. SEED RL: Scalable and efficient deep-rl with accelerated central inference. arXiv preprint arXiv:1910.06591, 2019.
> > [8] Fabian Abel, Yashar Deldjoo, Mehdi Elahi, and Daniel Kohlsdorf. RecSys Challenge 2017: Offline and Online Evaluation. In ACM Conference on Recommender Systems, 2017.
> > [9] Chow, Y., Nachum, O., Duenez-Guzman, E., & Ghavamzadeh, M. (2018). A lyapunov-based approach to safe reinforcement learning. In Advances in neural information processing systems (pp. 8092-8101).
> > ```

---

### Official Review · AnonReviewer2 · 2020-11-02

**Rating:** 7
**Confidence:** 4

**Review:**

The authors  highlight the problem of iterated batch RL (and thus its deployment efficency) and propose an algorithm along
with novel evaluation schemes.

There are several things I like about this paper:

- The topic is very important and underexplored, especially the deployment efficency w.r.t. the number of batch collections
- Lots of relevant work is cited
- The experiment make sense given the research question. Especially evaluations such as in Figure 2.
- The results show significant improvements of the proposed method over state of the art
- The method  is "simple", in a good way, and has not that much moving pieces, compared to other RL algorithms

The are a few things I think could be improved:

- Because the paper considers "repeated batch" it should touch the subject of "if I can collect x batches, what batches would tell me the most?". I.e the algorithm as written right now is "greedy" in the sense that the policy will only act in the way it considers most optimal (and is not completely different from the behavior policy).  Batch exploration strategies would be highly interesting
and as "iterative batch RL" is the domain of this paper, these need to be addressed.

- Related:  While epistemic uncertainty is modeled (via Ensembles of dynamic models) it is not explicitly utilized.  Eq. (4) contains a "safety mechanism" via the trust region approach.  Why not include a risk criterion over the ensemble spread?

- Why use only deterministic models?  Using something like [1,2] that enables modeling stochastic effects would improve the
universality of the approach further, not being restricted to deterministic problems.

I am a bit unsure about the novelty of Section 4  --  most pieces of the algorithm are already known or "obvious".  To me the
biggest novelty lies in identifying the "repeated batch" as  an important problem, that is underexplored and shows how
classical RL method underperform strongly here. From this  evaluations like shown in Figure 2 are novel and make a lot of sense.


[1] Chua, Kurtland, et al. "Deep reinforcement learning in a handful of trials using probabilistic dynamics models." Advances in Neural Information Processing

[2] Lakshminarayanan, Balaji, Alexander Pritzel, and Charles Blundell. "Simple and scalable predictive uncertainty estimation using deep ensembles." Advances in neural information processing systems. 2017.

---

> ### Author Response · Authors · 2020-11-17
> **Author Response to AnonReviewer2**
>
> We thank the reviewer for the careful reading of the paper. We are glad that the reviewer appreciates the motivation and methodology of the paper.
>
> **> Batch exploration method**
> Thank you for this suggestion! Indeed, “how to collect X batches” beyond our simple greedy + Gaussian-noise exploration heuristic (See Appendix D.2.1 “Stationary noise” and noise selection ablation in Appendix E.1) is an exciting future direction of research. We are actively exploring other heuristics, taking inspirations from acquisition functions from Bayesian optimization literature, as well as intrinsic motivation literature in RL [1, 2, 3], to our iterative batch RL settings for improved sample-efficiency.
>
> **> Utilization of epistemic uncertainty**
> In Appendix C, we do compare the variants of BREMEN incorporating soft or hard reward penalties with ensemble variances as proposed in the concurrent works MOReL [4] and MOPO [5], instead of BREMEN’s regularization with model uncertainty by sampling from ensembles. We found empirically that BREMEN performs consistently better than these variants, which seem overly pessimistic in the limited deployment-settings.
>
> **> Stochastic models**
> As the reviewer pointed out, our proposed method uses an ensemble of deterministic models, and we agree using a probabilistic model for managing stochasticity of the dynamics is a nice extension that can further strengthen BREMEN performance. Thank you for providing us with the references!
>
> **> Novelty of Section 4**
> Indeed, many components of our proposed algorithm are borrowed from previous literature. The key contributions of our paper are (1) the proposal/evaluation of deployment-efficiency metric, (2) the proposal/evaluation of sample-efficient offline RL benchmarks (20x less data), and (3) the development of a simple algorithm for (1)+(2) through a novel combination of existing ideas (model-ensemble+trust-region from ME-TRPO, and BC initialization from offline literatures).
>
> ```
> [1] Rein Houthooft, Xi Chen, Yan Duan, John Schulman, Filip De Turck, and Pieter Abbeel. VIME: Variational Information Maximizing Exploration.  In Advances in Neural Information Processing Systems, 2016.
> [2] Deepak Pathak, Dhiraj Gandhi, and Abhinav Gupta. Self-Supervised Exploration via Disagreement. In International Conference on Machine Learning, 2019.
> [3] Burda, Yuri, et al. "Exploration by random network distillation." arXiv preprint arXiv:1810.12894 (2018).
> [4] Rahul Kidambi, Aravind Rajeswaran, Praneeth Netrapalli, and Thorsten Joachims. MOReL: Model-based offline reinforcement learning. arXiv preprint arXiv:2005.05951, 2020.
> [5] Tianhe Yu, Garrett Thomas, Lantao Yu, Stefano Ermon, James Zou, Sergey Levine, Chelsea Finn, and Tengyu Ma. MOPO: Model-based offline policy optimization. arXiv preprint arXiv:2005.13239, 2020.
> ```

---

### Author Response · Authors · 2020-11-24
**Summary of Revisions**

We would like to thank all the reviewers for the insightful comments, which helped us improve the paper. We have updated the manuscript according to the constructive suggestions from the reviewers.

The summary of major update made to the paper is as follows (highlighted with purple letters in the manuscript):
1. Modified the conclusion section (section 7) in order to clarify the contribution of the paper.
2. Added results of experiments on the trade-off of sample efficiency and deployment efficiency in Appendix B.
3. Added discussion about the importance of deployment efficiency in real world applications in Appendix C.

We have also made tiny fixes on typos.

---

### Decision · Program_Chairs · 2021-01-07
**Final Decision**

**Decision:**

Accept (Poster)

**Comment:**

# Quality:
The algorithm is thoroughly evaluated and several interesting experiments are included in the appendix.

# Clarity:
The paper is generally well written.

# Originality:
The proposed approach is a small but novel improvement over existing algorithms (to the best of the reviewers and my knowledge). The concept of "deployment-efficiency" is, in my opinion not novel, since it seems mostly a rebranding of what the MBRL community traditionally refers to as "data-efficiency" -- although I agree that deployment-efficiency is indeed a more accurate term.

# Significance of this work:
The paper deal with a relevant and timely topic. However, the paper does not compare to the larger MBRL literature. Hence, it is difficult to gauge the significance of this work.

# Overall:
This manuscript offers a good contribution to the topic of model-based reinforcement learning algorithms.

# Minor comments:
- I suggest removing the word "impressive" from the abstract. This is a subjective term, which should be avoided.
- In my personal opinion, it would be nice to include experiments with more state-of-the-art baselines such as PETS and POPLIN, for which code is available online. It is unclear to me how much the improvement in performance depends on the algorithm itself compared to just having larger batch sizes. From this perspective, Figure 5 in Appendix B is probably the most interesting insight of the manuscript, to me.